



# Seasonal evolution of basal environment conditions of Russell sector, West Greenland, inverted from satellite observation of surface flow

Anna Derkacheva[1,*], Fabien Gillet-Chaulet[1,*], Jeremie Mouginot[1,2,*], Eliot Jager[1], Nathan Maier[1], and Samuel Cook[1]

[1]Université Grenoble Alpes, CNRS, IRD, INP, 38400 Grenoble, Isère, France
[2]Department of Earth System Science, University of California, Irvine, 92697 CA, USA
[*]These authors contributed equally to this work.

**Correspondence:** Anna Derkacheva (anna.derkacheva@univ-grenoble-alpes.fr, der_a@mail.ru)

**Abstract.** Increasing surface melting on the Greenland ice sheet requires better constraints on seasonally evolving basal water pressure and sliding speed. Here we assess the potential of using inverse methods on a dense time series of surface speeds to recover the seasonal evolution of the basal conditions in a well-documented region in southwest Greenland. Using data compiled from multiple satellite missions, we document seasonally evolving surface velocities with a temporal resolution of
two weeks. We then apply the inverse control method using Elmer/Ice to infer the basal sliding and friction corresponding to each of the 24 surface-velocity data sets. Near the margin where the uncertainty in the velocity and bed topography are small, we obtain clear seasonal variations that can be mostly interpreted in terms of a effective-pressure based hard-bed friction law. We find for valley bottoms or "troughs" in the bed topography, the changes in basal conditions directly respond to local water pressure variations, while the link is more complex for subglacial "ridges" which are often non-locally forced. At the catchment
scale, in-phase variations of the water pressure, surface velocities, surface-runoff variations are found.Our results show that time-series inversions of observed surface velocities can be used to understand the evolution of basal conditions over different timescales and could therefore serve as an intermediate validation for subglacial hydrology models to achieve better coupling with ice-flow models.

## 1   Introduction

In recent decades, the Greenland ice sheet (GrIS) has been constantly losing mass, reaching a negative mass balance of about $-286 \pm 20$ Gt/y in 2010–2018 (Mouginot et al., 2019). An important part of this loss comes from the ice-discharge acceleration of tidewater glaciers. In addition, ice dynamics also plays a significant role in land-terminating sectors as ice flow affects the flux of ice into the ablation area and the ice-sheet topography, with feedbacks between ice-sheet surface elevation and the atmosphere that can enhance the mass loss in century-scale projections (Edwards et al., 2014; Le Clec'h et al., 2019).

Water pressure at the glacier base is considered to be a major control on basal sliding that affects ice dynamics over different timescales (Nienow et al., 2017; Davison et al., 2019). For instance, seasonal modulations of water input to the bed, induced by summer melt, are able to lead to peak glacier accelerations of up to +360 % locally compared to winter mean velocities (Palmer et al., 2011). Increased water pressure as melt drains through an inefficient drainage system is assumed to be the mechanism





driving the acceleration during the beginning of the melt season. However, increased drainage efficiency during the late melt

season leads to a decrease in water pressures and causes a commensurate glacier deceleration. Moreover, during high melt years, higher early summer velocities are therefore posited as responsible for the slower velocities during the late melt and winter seasons, offsetting the higher initial ice flux (Tedstone et al., 2015). This suggests that the future of the GrIS will then be affected by the evolution of the surface runoff discharge and its effect on the subglacial drainage system.

The rate of surface melting is already increasing due to warming of the air, as well as other factors amplifying this phe-

nomenon such as the decrease in the ice albedo (Box et al., 2012), decrease in the capacity of the firn to retain melt water (Mikkelsen et al., 2016), or even change in the dominant weather type (Van Tricht et al., 2016). Climate models predict that this melting will continue to grow in the future. The surface runoff is also increasing both in observations and projection models (Ahlstrøm et al., 2017; Trusel et al., 2018). However, it is still debated how the ice-flow velocity will respond to this enhancement in water production in the long term; either with a continuous increase (Zwally et al., 2002; Greskowiak, 2014;

Hewitt, 2013), an increase until a threshold (Tedstone et al., 2013, 2014; Poinar et al., 2015) or even a decrease (Tedstone et al., 2015; Stevens et al., 2016). A complete analysis of all these hypotheses suggests that different trends will dominate according to the timescale and altitudes considered (Nienow et al., 2017; Davison et al., 2019).

These questions have motivated the development of physical models to represent the subglacial environment and its interaction with the ice dynamics. Basic ingredients of these models are (i) a subglacial hydrological model that computes the effective

pressure and (ii) a friction law that relates the basal shear stress to the effective pressure and the basal sliding velocity. However, because of the limited accessibility, the processes at the bed remain difficult to characterize, limiting their understanding. Both components are still a matter of debate and no consensus has emerged.

Flowers (2015) gives a review of available subglacial hydrology models and their theoretical background. Thirteen models of various complexity have participated in a recent model-intercomparison exercise that shows that physical approaches cou-

pling several elements of the basal drainage system significantly differ from simpler approaches for short term, e.g. diurnal, variations (De Fleurian et al., 2018). To be run operationally these physical models require highly detailed input data (e.g. basal topography, runoff forcing) that are often not available, and they suffer from a lack of direct and independent data for calibration and validation. The most direct way to access the basal hydrology system is to drill boreholes that allow direct measurements of the water pressure (Smeets et al., 2012; Meierbachtol et al., 2013; Van De Wal et al., 2015; Wright et al.,

2016). While very valuable, these local measurements can show a high spatial variability depending on the element of the basal hydrology that is sampled (Wright et al., 2016), so these observations make the validation of subglacial hydrology models challenging, as they are not necessarily representative of the large-scale average basal conditions that are required to reproduce and predict the long-term evolution of the entire glacier. However, basal hydrology models have been applied in real settings, and the comparison with available observations has stimulated their development. For instance, in de Fleurian et al. (2016)

the timing of the response of modelled water pressure broadly agrees with observations but with a significant difference in terms of magnitude. In addition to an efficient and inefficient drainage system, Hoffman et al. (2016) introduced a third weakly connected component to explain the decline of water pressure during the late melt season. Alternatively, Downs et al. (2018) suggest a reduction in hydraulic conductivity to explain the tendency of models to underpredict observed winter water pressure.





The second required component, the friction law, depends on the properties of the bed. Deformable basal sediments (com-
monly referred to as soft or weak beds) are usually modelled using a Mohr-Coulomb criterion to relate the basal shear stress to
the effective pressure (Iverson et al., 1998; Fowler, 2003; Joughin et al., 2019; Helanow et al., 2021). For hard beds (rigid rocks
or non-deformable till, as opposed to deformable till), the friction is controlled by the ice deformation over the small-scale
basal roughness, inducing a relationship between the basal shear stress and the sliding velocity. Increasing the water pressure
can open subglacial cavities, reducing the apparent bed roughness and the basal friction (also referred to as basal shear stress or
basal traction). Several friction laws that incorporate the dependency on the effective pressure have thus been developed both
from theoretical (Schoof, 2005) and empirical considerations (Budd et al., 1984). As for the subglacial hydrology, because
of the inaccessibility of the basal environment, the bed properties of specific glaciers are generally poorly known. Addition-
ally, geophysical investigations have shown evidence for the presence of deformable sediments and hard beds in relatively
close proximity (Dow et al., 2013; Harper et al., 2017). Thus, in-situ direct validation of the friction law is not possible, so
that the models must be evaluated against surface velocities, necessarily inducing uncertainties in the basal hydrology and ice
deformation.

Synthetic simulations of typical Greenlandic land-terminating glaciers using coupled basal-hydrology-ice-dynamics models
have been able to reproduce the main observed features of the seasonal modulation of surface velocity (Hewitt, 2013; Gagliar-
dini and Werder, 2018; Cook et al., 2020). In real applications, models have mostly been validated using velocity fluctuations
recorded by GPS at the ice-sheet surface (Bougamont et al., 2014; Kulessa et al., 2017; Christoffersen et al., 2018; Koziol and
Arnold, 2018). Again, being precise, these measurements are local and do not allow the spatial variability of the processes to
be properly constrained.

Satellite imagery allows us to derive surface velocity fields with a good spatial resolution and coverage. During the last
decade, the number of such observations has increased significantly with the launches of missions such as Landsat-8 or
Sentinel-1&-2 (Fahnestock et al., 2016; Mouginot et al., 2017; Joughin et al., 2018; Lemos et al., 2018), allowing the re-
construction of flow variations at the seasonal scale with a temporal resolution of days to weeks (Altena and Kääb, 2017; Vijay
et al., 2019; Derkacheva et al., 2020).

Most recent ice-flow models are now equipped with various inverse methods that make it possible to spatially constrain
a free parameter that relates the basal friction to the sliding velocity using the observed geometry and surface velocity field
(MacAyeal, 1993; Arthern and Gudmundsson, 2010). Several studies have then tried to validate or constrain the friction law
from the inferred basal friction and velocity. As the velocities in ice sheets can range over several orders of magnitude, this can
be assessed from the spatial variations in a single inversion, assuming that changes are mostly driven by the friction law and
not by spatial variations of the bed properties. Thus, Arthern et al. (2015) found that the basal stress in Antarctica, on average,
roughly agrees with a uniform value of 100 kPa, however this can change locally by order of magnitudes. Spatially aggregating
inversions with models of different complexity, Maier et al. (2021) found that large areas under the Greenland ice sheet broadly
agree with hard-bed physics. The other possibility to constrain the friction law is to use several inversions to study the temporal
changes; however this can be done only where the changes are sufficiently large. For instance, Gillet-Chaulet et al. (2016)





found that changes in the drainage basin of Pine Island Glacier in Antarctica over a 14-year period can be explained with a mostly plastic relation, where the basal friction is weakly sensitive to changes in sliding velocity.

At the same time, for areas that can be affected by temporal variations of the basal hydrology, several studies used the inferred velocity fields to constrain temporal variations of the basal water pressure by assuming a certain friction law that also includes the effective pressure (Jay-Allemand et al., 2011; Minchew et al., 2016). Thus, the main objective of this paper is to assess the ability of existing inverse methods to use satellite-derived seasonal velocity maps to infer seasonal variations in the basal conditions.

We focus on a land-terminating sector of the southwest coast of Greenland located at 67°N 50°E. This sector includes a slow moving ice-sheet margin and three distinct glaciers (from north to south): Insunnguata Sermia, Russell Gletscher, Ørkendalen Gletscher (Fig.1). Hereafter, this area is referred to as Russell sector. Extensive measurements of the ice thickness were carried out over the studied region using radar sounders, especially through NASA's Operation IceBridge mission (Morlighem et al., 2013; Lindbäck et al., 2014). This dense dataset with an average radar-line spacing of less than 500 m is exceptional by 105 Greenland standards, where most radar lines are generally separated by tens of kilometers. In addition, because of its relative accessibility, this sector has been the subject of numerous complementary geophysical investigations such as boreholes, seismometers, or GPS (Smeets et al., 2012; Dow et al., 2013; Wright et al., 2016; Harper et al., 2017; Kulessa et al., 2017; Maier et al., 2019), making it a privileged study site for numerical investigations (Bougamont et al., 2014; de Fleurian et al., 2016; Koziol and Arnold, 2017, 2018; Downs et al., 2018; Christoffersen et al., 2018; Brinkerhoff et al., 2021).

We use the three-dimensional finite-element full-Stokes ice-flow model Elmer/Ice (Gagliardini et al., 2013) to invert for the seasonal evolution of the basal friction and sliding speed using surface velocity maps covering an entire year with a time step of two weeks. We address the questions of how best to integrate satellite-derived velocity into a model, as well as the sensitivity of the inverted basal friction fields to initial ice temperature and some commonly used model parameters. From the inverted basal fields, we estimate the corresponding evolution of the water pressure using a pressure-dependent friction law (Schoof, 2005; 115 Gagliardini et al., 2007). Results are discussed using available in-situ measurements and outputs from numerical models of subglacial hydrology and regional climate. Finally, we conclude on the usability of inverse-flow models with spatio-temporally dense observations of surface velocity to derive the seasonal evolution of the glacier basal environment.

## 2   Seasonal surface velocities

We have derived a time series of horizontal surface ice velocity of the Russell sector at a spatial resolution of 150 m using 120 satellite images collected between 2015 and 2019 by Landsat-8, Sentinel-1, and Sentinel-2. The details on the data processing can be found in Derkacheva et al. (2020), and are summarized below.

    A cross-correlation approach is used to estimate the features (or radar speckle) displacement between master and slave images taken on two different dates, which is further converted to the $v_x$ and $v_y$ surface flow speed components (Mouginot et al., 2017; Millan et al., 2019). Only the measurements with time intervals shorter than 32 days (1 month) are used in order 125 to capture rapid dynamic changes in ice flow. To reduce the noise and relatively large errors associated with these short revisit


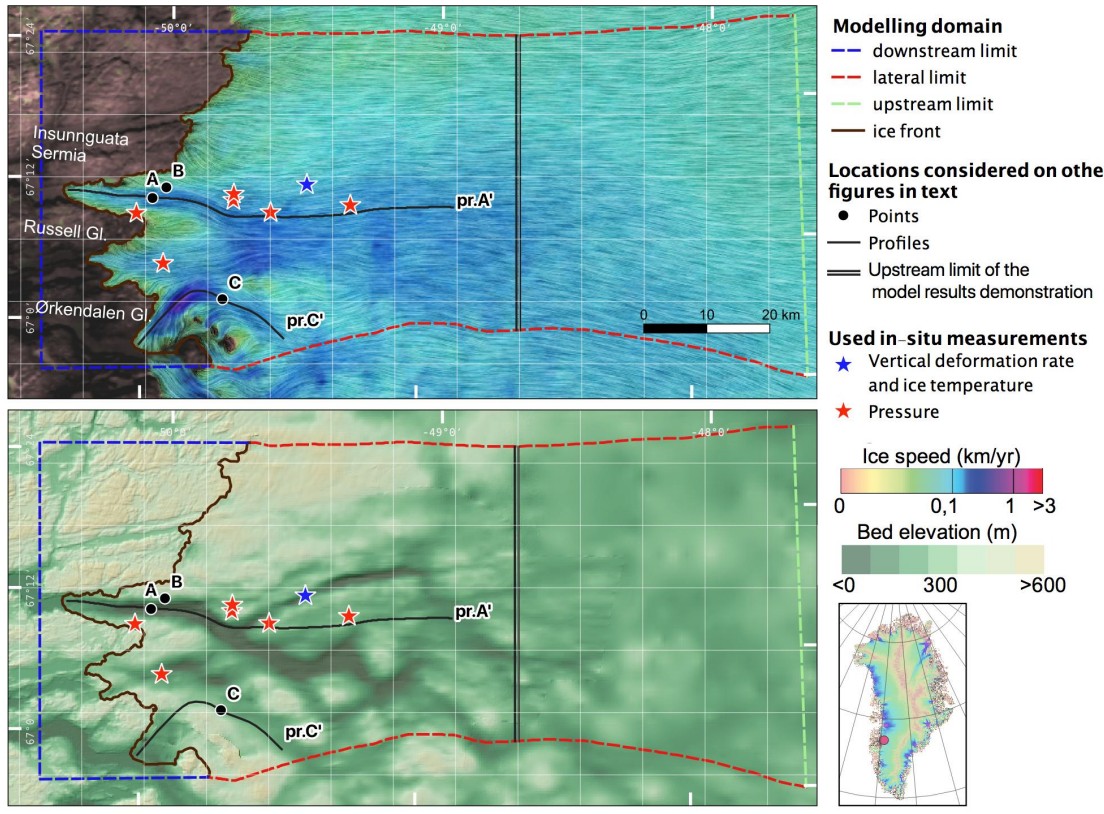

**Figure 1.** Study area with modelling domain. The points (A, B, C) and profiles (pr. A', pr. C') indicate locations considered later in the text, and the blue and red stars indicate locations of in-situ measurements (Smeets et al., 2012; Meierbachtol et al., 2013; Wright et al., 2016; Hills et al., 2017; Maier et al., 2019). The top panel shows the surface ice velocity overlaid on the line convolution integral to visualize the flow vector direction (Mouginot et al., 2017). The bottom panel displays the bed elevation from BedMachine (Morlighem et al., 2017). The white 10km grid used here is identical in all figures in the article. The projection used is Polar Stereographic North with latitude of true scale at 70°N.

times, a linear non-parametric local regression (LOWESS) smoothing algorithm is applied on the resulting time series. The final accuracy of the product has been validated against in-situ GPS measurements from Maier et al. (2019) showing that they can be used to describe ice-velocity fluctuations with a temporal resolution of about 2 weeks over a large area.

To further minimize the noise and increase the spatial coverage we also reduce this five-year time series to a "typical year". To do so, we compute the median value for a given calendar interval over all five years simultaneously, averaging on a regular step of a half-month, referred to hereafter as "early" and "late" halves. We thus obtain a final product of 24 maps describing the median behaviour of annual ice-flow variations. The two surface velocity components $vx$ and $vy$ for early July are shown in Fig.2-a and -b. This temporal aggregation is justified by the fact that, between 2015 and 2019, this sector experienced a relatively similar pattern of speed variations each year without outstanding extremes, thus the median year



is fairly representative of the behavior of this sector during the considered period. For instance, we compare in Fig.2-f the temporal evolution of the median speed with the annual data sets at a location on Insunnguata Sermia (point A on Fig.1). The root-mean-squared deviation between the median and the annual data sets is about 10 m/yr, which is within 10 % of the mean winter speed. In addition to the bi-monthly time series, we also averaged early and late January, February and March observations to produce a map of Mean Winter Speed (MWS hereinafter).

Besides the velocity maps, we generate a spatially distributed estimation of the uncertainty for each temporal step. This uncertainty $\sigma$ is computed per pixel as $\sigma = STD/\sqrt{n}$ where $STD$ is the standard deviation of the averaged measurement in each time step and $n$ is the number of averaged measurements. Due to the varying characteristics of the sensors and the resulting observations, the accuracy of the generated velocity maps varies over the time and space. While for winter maps (January-April, December), the errors are generally below 10 m/yr, these values can rise three-to-fourfold between May and
November, especially for the most inland areas where fewer observations are available and the smooth terrain makes satellite speed tracking difficult (see Fig.2-c and -d and Fig.A1).

       Our averaged data set is consistent with previous satellite products and ground observations in the area (Joughin et al., 2008; Palmer et al., 2011; Fitzpatrick et al., 2013; Lemos et al., 2018), as we observe the same range of speed magnitudes and seasonal changes. The average winter speed varies from 50 m/yr outside of the main glacier trunks, to 100-250 m/yr at
Insunnguata Sermia and Russell Gletscher, and up to 300 m/yr at Ørkendalen Gletscher. At the end of spring and into summer, a pronounced speed-up is observed for the entire sector, with the acceleration starting at the ice margin and gradually moving upstream. As an extreme case, a short-term acceleration up to +360 % above MWS was observed over one small area (Palmer et al., 2011). However, the mean range of the speed acceleration spatially varies between +100 % to +250 % above MWS. Depending on the location, the deceleration starts in late June or July, continuing for one to four months. Consequently, the
autumn velocity can be lower than the winter mean, which is especially typical for Ørkendalen Gletscher.

       The ice in this sector flows in a clear east-west orientation with an averaged flow direction of about 275°from the North, except for Ørkendalen Gletscher. Therefore, the y-velocity component ($v_y$) is small (Fig.2-b), and small variations in it can induce relatively large changes in the estimated flow direction. We noted in our time series that the flow direction varies up to $\pm25°$across the year. This effect is only observed in velocity fields derived from the optical images and not in the
synthetic aperture radar images, suggesting that this phenomenon is not related to a real change in ice-flow direction. We attribute this effect to changes in surface illumination and shadow length being a function of solar elevation change from March to October. This is assumed to induce the supplementary displacements of the surface-feature footprints tracked by cross-correlation algorithms. Indeed, on Insunnguata Sermia, the mean $v_y$ speed vector component changes from +60 m/yr to +10 m/yr and -40 m/yr between the velocity fields derived with Sentinel-2 optical images in early March (ascending low sun),
late June (solstice) and early September (descending low sun) correspondingly. To deal with this phenomenon, we assume that the total magnitude of our time series was not severely affected and so we only consider the norm of the velocity vector for the model inversions, keeping the vector direction to be defined by the model itself.

       According to theoretical expectations, the ice flow is most likely to be affected by seasonal variations in surface runoff reaching the bed across the area that is under the equilibrium line. At Russell, the long-term equilibrium line is estimated to be





at 1500 m (Van De Wal et al., 2012). Even so, the GPS records have shown the presence of a short summer speed-up further inland as well (Bartholomew et al., 2012; Greskowiak, 2014), with acceleration of about +5 % up to 50 km inland from the equilibrium line (Greskowiak, 2014). That corresponds to about +1 m/yr above the local MWS and is within the noise level in our velocity dataset. For this reason the area of interest is limited to approximately 100 km upstream from the ice margin, which corresponds to a surface elevation of about 1400 m.

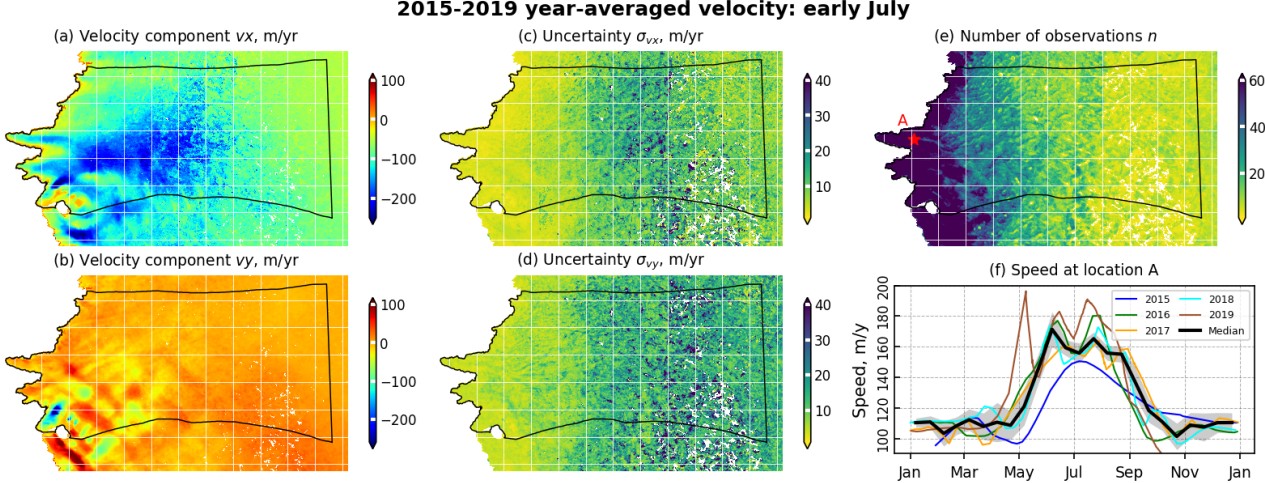

**Figure 2.** Year-averaged velocity dataset. (a) - (e) Horizontal surface velocity components ($v_x$ and $v_y$), associated uncertainties ($\sigma_{v_x}$ and $\sigma_{v_y}$), and number of observations per pixel ($n$). (f) Comparison between the original speed observations for years 2015 to 2019 and the year-averaged data set where the grey shading represent the 1-$\sigma$ interval at point A in (e) and in Fig. 1. White pixels on maps (a), (b), (e) correspond to areas that are ice-free or with no data, and on maps (c), (d) to pixels with less than 2 velocity observations.

## 175 3 Methods and model description

We use the finite-element ice-flow model Elmer/Ice (Gagliardini et al., 2013) to compute the 3D velocity field in the Russell basin and invert it for the basal conditions over an entire year using the 24 surface velocity maps.

We use the basal topography given by BedMachine version 3 (Morlighem et al., 2017). The surface elevation comes from the Greenland Ice Mapping Project (GIMP, Howat et al., 2014) which has a nominal date of 2007. This is 10 years earlier

than our velocity observations. Thickness changes in this area are about -1 m/yr near the margin (Helm et al., 2014), which is relatively small compared to the ice thickness (see Fig.A2-b) thereby we use it as it is. It also seems reasonable to assume that the topography does not change significantly over a single year, and we therefore keep the surface topography fixed for all inversions. In doing so, we assume that basal changes are the only drivers of the seasonal ice-speed variations and that changes in driving stresses due to surface-elevation variations are negligible.

In this section we describe the modelling domain, the model set-up and the inverse modelling method.



## 3.1 Modelling domain and mesh

Our model domain corresponds to the ice-catchment basin of the three land-terminating glaciers mentioned above, reaching to about 100 km inland for the reasons explained in Sec.2 (Fig.1). The lateral borders of the domain are defined by flow lines derived from the mean multi-annual observed surface velocity (Rignot and Mouginot, 2012). As the margin is land-terminating

and thus does not require special boundary conditions at the front, the model domain has been extended a few kilometers in front of the margin in expectation of future transient simulations, leaving the possibility of an advance of the glaciers open.

To create the mesh, we start by meshing the horizontal footprint of the domain using the Elmer/Ice anisotropic mesh capabilities that rely on the MMG library (https://www.mmgtools.org/, Dapogny et al. (2014)). The mesh adaptation scheme equi-distributes the interpolation error of the observed surface velocities and thickness. Because the domain is relatively small,

the mesh resolution is allowed to vary between 150 m near the margin and 400 m up to 40 km inland, with a maximum value of 1 km attained progressively beyond this. The resulting two-dimensional irregular mesh consists of approximately 60'000 linear triangular elements and approximately 30'000 nodes. It is then vertically extruded into 20 layers to form the 3D mesh. As the vertical velocity gradients are expected to be higher near the bed, the vertical resolution increases following a geometric progression and is two times smaller at the bed than at the surface. The 3D mesh is then stretched vertically so that the top and

bottom surface follow the topographies given by GIMP and BedMachine. As the mesh cannot have a null thickness, the upper surface is corrected to give a minimum thickness of 0.9 m at the ice-free areas.

Note that our study area is well-constrained by the radar measurements of ice thickness in the lower half of the domain, while the upper part of the basin has been much less surveyed (see Figure A2-a). A mass-conservation algorithm Morlighem et al. (2011) has been used for the interpolation of the basal topography in the densely surveyed area, while kriging is used

outside. Reported uncertainties on the bed elevation over the lower part of our domain, except the steep-slope front line, are generally lower than 30 m or below 6 % of ice thickness, while on the upper part they can reach up to 300 m, or 20 % of the ice thickness. We also notice that a few small-scale features are not well captured in the BedMachine version used here (v3.10, 20-Sep-2017), such as the south-western nunataks next to Ørkendalen Gletscher that are actually ice-free but for which the data displays an ice thickness of 100-140 m. As for the surface data provided by GIMP, they are constructed from a combination

of ASTER and SPOT-5 DEMs for the ice-sheet periphery and margin, and AVHRR photoclinometry in the ice-sheet interior (Scambos and Haran, 2002). Their uncertainty has been estimated by comparing with spaceborne lidar altimetry from ICESat, and is about ±1 m over most areas of interest (Howat et al., 2014).





## 3.2 Direct model

### 3.2.1 Field equations

To compute the 3D ice velocity ($\boldsymbol{u}$) and ice pressure ($p_i$) fields we solve the Stokes equations that express the conservation of momentum and mass for an incompressible fluid:

$$\begin{cases} \operatorname{div}(\boldsymbol{\sigma}) + \rho\boldsymbol{g} = 0 \\ \operatorname{div}(\boldsymbol{u}) = 0 \end{cases} \tag{1}$$

where $\rho$ is the ice density, $\boldsymbol{g}$ is the gravity vector and $\boldsymbol{\sigma} = \boldsymbol{\tau} - p_i\boldsymbol{I}$ is the Cauchy stress tensor with $\boldsymbol{\tau}$ the deviatoric stress tensor and $\boldsymbol{I}$ the identity matrix.

To close the system, we use the classical viscous isotropic power law, known as Glen's flow law, that non-linearly relates the deviatoric stress tensor to the strain-rate tensor $\dot{\boldsymbol{\epsilon}}$ as:

$$\boldsymbol{\tau} = 2\eta\dot{\boldsymbol{\epsilon}} \tag{2}$$

where the effective ice viscosity $\eta$ is given by

$$\eta = \frac{1}{2}\left(EA\right)^{-1/n}\dot{\epsilon}_e^{(1-n)/n} \tag{3}$$

where the second invariant of the strain-rate tensor is given by $\dot{\epsilon}_e^2 = \operatorname{tr}(\dot{\boldsymbol{\epsilon}}^2)/2$, $n$ is the Glen exponent, $E$ is an enhancement factor and the rate factor $A$ depends on the ice temperature $T$ following an Arrhenius relationship:

$$A = A_0 e^{-\frac{Q}{RT'}} \tag{4}$$

where $A_0$ is the pre-exponential factor, $Q$ is an activation energy, $R$ is the gas constant, and $T' = T - T_p$ where the pressure melting point is given by $T_p = 273.15 - C_c p_i$ with $C_c$ the Clausius-Clapeyron constant.

Initialising the temperature field is a difficult problem as the thermal-state in an ice sheet has a long-term memory requiring multi-millennial spin-up simulations (Goelzer et al., 2017), and the heat sources are in general poorly constrained and make the thermo-mechanical problem non-linear (Schäfer et al., 2014). Here we use the temperature field simulated by the ice-sheet model SICOPOLIS for the present state of the Greenland ice sheet (Goelzer et al., 2020).

We have compared the temperature profiles to existing in-situ borehole measurements done in the ablation zones of Insunnguata Sermia (Harrington et al., 2015; Hills et al., 2017). Results are shown in supplementary Fig.A3. The modelled temperature appears to fit the observations better at higher altitudes around 40 km from the ice margin (Hills et al. (2017); Harrington et al. (2015), location S5) than at lower altitudes (Harrington et al. (2015), locations S2-S4), certainly because SICOPOLIS does not have the resolution and processes to accurately capture the individual land-terminating glaciers. Within the first 40 km from the glacier terminus, observed temperatures are generally warmer than $-6°$C across the entire ice column, while the model finds the temperature to be up to $6°$C cooler across the ice column. Further inland from the glacier terminus, both measurement campaigns found that the temperature decreases from -10° at the surface to $-13°$C at a depth of



**Table 1.** The list of the constants used in the model.

| Description | Value | Units |
| --- | --- | --- |
| Gravity constant $g$ | 9.8 | m s$^{-2}$ |
| Ice density $\rho$ | 910 | kg m$^{-1}$ |
| Glen exponent $n$ | 3 | |
| Pre-exponential factor $A_0$ | $2.84678 \times 10^{-13}$ for $T' < 10°$C | Pa$^{-3}$s$^{-1}$ |
| | $2.35567 \times 10^{-2}$ for $T' \geq -10°$C | |
| Activation energy $Q$ | 60 for $T' < 10°$C | kJ mol$^{-1}$ |
| | 115 for $T' \geq -10°$C | |
| Gas Constant $R$ | 8.314 | J K$^{-1}$ mol$^{-1}$ |
| Clausius-Clapeyron constant $C_c$ | $9.8 \times 10^{-2}$ | K MPa$^{-1}$ |
| Enhancement factor E | 1 | |

about 200-300 m, and then rises to near-melting temperature at the glacier base. Here the SICOPOLIS temperature follows a similar trend with a slight $1-2°$C warmer divergence over the ice column, but becomes cooler about 200 m above the bedrock. At the glacier base, the model shows in all boreholes a deviation of $2-4°$C below the measurement, meaning that

the measured temperature at the base is closer to the melting point than SICOPOLIS estimates. As deformation rates increase rapidly and non-linearly with increasing ice temperature (Eq.4), the generally colder modelled ice could potentially cause the ice-flow model to underestimate the internal deformation compared to reality. However, the investigation of the influence of temperature-field uncertainties on the basal-friction inversions show a more limited influence than, for instance, uncertainties in the basal topography (Habermann et al., 2017). As the temperature field reproduces sufficiently well part of the observed

spatial variability, in the following, we briefly assess the sensitivity of the results to the ice rheology only by changing the value of the enhancement factor $E$: to validate the final rheology parameterisation, the model-derived ice-deformation profiles are compared further against in-situ borehole measurements (Sec.4.3).

### 3.2.2 Boundary condition

As we model only a part of the ice sheet, in addition to the conditions at the top and bottom surfaces we have to prescribe

boundary conditions at the sides of the domain. Note that the lateral sides of the domain have been chosen to be sufficiently far from the regions of interest so that, for the diagnostic simulations presented here, the details of the boundary conditions should not affect the solution at distances higher than a few ice thicknesses. For all the boundaries, we denote by $n$ the unit vector normal to the boundary and pointing outward from the model domain.





The lateral sides of the domain coincide with flow lines so there should be no ice flux entering the domain. We therefore impose the following condition:

$$\boldsymbol{u}.\boldsymbol{n} = \boldsymbol{0} \tag{5}$$

Along the tangential directions, we keep the natural stress-free condition and thus neglect the tangential shearing components along these boundaries. At the inflow boundary ($\Gamma_i$) we apply Dirichlet conditions for the horizontal components of the velocity vector ($\boldsymbol{u}_H = (u_x, u_y)$) using the observed surface winter-mean velocities ($\boldsymbol{u}_s^{obs}$) (see Sec.2):

$$\boldsymbol{u}_H = \boldsymbol{u}_s^{obs} \tag{6}$$

Note that at the start we impose uniform velocities along the vertical direction and thus neglect the deformation profile. This is consistent with observations in the area (Maier et al., 2019) and with our model results (Sec.4.3), which show that the deformation profiles contribute to only a small portion of the surface velocities. We leave the natural stress-free condition for the vertical direction and thus neglect vertical shearing along this boundary.

The bottom and top surfaces correspond to natural interfaces of the ice. For the upper surface, $\Gamma_s$, we neglect the atmospheric pressure and impose a stress-free condition:

$$\boldsymbol{\sigma}.\boldsymbol{n} = 0 \tag{7}$$

For the bottom boundary, $\Gamma_b$, we use a linear friction law that relates the tangential basal shear stress, $\boldsymbol{\tau}_b = \boldsymbol{T}.\boldsymbol{\sigma}.\boldsymbol{n}$, to the basal sliding velocity $\boldsymbol{u}_b = \boldsymbol{T}.\boldsymbol{u}$, and a no-penetration condition for the normal velocity:

$$\begin{cases} \boldsymbol{\tau}_b + \beta \boldsymbol{u}_b = 0 \\ \boldsymbol{u}.\boldsymbol{n} = 0 \end{cases} \tag{8}$$

where $\boldsymbol{T} = \boldsymbol{I} - \boldsymbol{n} \otimes \boldsymbol{n}$ is the tangential operator, $\beta$ is an effective friction coefficient which is tuned using the inverse procedure described in the following section. Hereafter, we will mainly refer to the norm of of the sliding velocity and basal friction denoted by $u_b = |\boldsymbol{u}_b|$ and $\tau_b = |\boldsymbol{\tau_b}|$.

The results of the inversions using a fixed geometry and a single velocity data set have been shown to be weakly sensitive to the choice of the friction law as the friction field must satisfy the global stress balance (Joughin et al., 2004). This has been confirmed in the area by Koziol and Arnold (2017), who found very small differences when comparing the basal shear stress fields inverted using three different friction laws.

### 3.3 Inverse model

Inverting the basal friction coefficient using the observed surface velocities is now widespread in many ice-sheet models. Here, we use the variational control inverse method implemented in Elmer/Ice (Gagliardini et al., 2013), and in the following we highlight the main steps.





For a given observed surface velocity field, the optimal effective basal friction field $\beta$ in Eq.8 is found by minimizing the following cost function

$$J_{tot} = J_0 + \lambda J_{reg} \tag{9}$$

where $J_0$ is an error term that measures the mismatch between model and observed surface velocities and $J_{reg}$ is a regularisation term weighted by the regularisation parameter $\lambda$. As the effective friction coefficient must remain positive, we use the following change of variable $\beta = 10^\alpha$ and the optimisation is done on $\alpha$.

We define $J_0$ as

$$J_0 = \frac{1}{2} \sum_1^{N^{obs}} \left( \frac{|\boldsymbol{u}_H^{mod}| - |\boldsymbol{u}_H^{obs}|}{|\boldsymbol{\sigma_u}|} \right)^2 \tag{10}$$

where $|\boldsymbol{u}_H^{mod}|$ is the norm of the model horizontal surface velocity vector interpolated at the $N^{obs}$ locations where we have an observation for the horizontal surface velocity $\boldsymbol{u}_H^{obs}$ (the nodes of the storage file grid, in fact), with $\boldsymbol{\sigma_u}$ the norm of the vector composed by the estimated velocity uncertainties (see Sec.2). Note that here we have chosen to compare only the norm of the vector and disregard the error on the flow direction, as the direction is mainly governed by the topography and, additionally, may be biased in our velocity observations, as explained in Sec.2.

To regularise the inverse problem, $J_{reg}$ is a Tikhonov regularisation term that penalises the horizontal derivatives of $\alpha$ as

$$J_{reg} = \frac{1}{2} \int_{\Gamma_b} \left( \frac{\mathrm{d}\alpha}{\mathrm{d}x} \right)^2 + \left( \frac{\mathrm{d}\alpha}{\mathrm{d}y} \right)^2 \, \mathrm{d}\gamma \tag{11}$$

The gradients of $J_{tot}$ with respect to the nodal values of $\alpha$ are computed by using the adjoint model and the cost function is minimized using the limited memory quasi-Newton routine M1QN3 (Gilbert and Lemaréchal, 1989).

## 4  Inferred basal sliding speed and friction

### 4.1  Mean winter state

#### 4.1.1  Model set-up

We start with an inversion of the MWS velocity map to get the initialisation state for our seasonal investigations. For that, the friction coefficient is initialised using a previous inversion performed with Elmer/Ice for the whole Greenland ice sheet using the shallow-shelf equations for the force balance and the minimisation is stopped after 200 evaluations of the cost function. Based on that, we run 6 inversions for different values of the regularisation parameter $\lambda$. Following the L-Curve method (Hansen, 2002), the optimal parameter is a compromise between fitting the observations and having a smooth solution which should correspond to the point of highest curvature in a log-log plot of the regularisation term against the error term. We plot those L-Curves and report the mean relative error $\bar{J}_0 = \sqrt{(2J_0/N^{obs})}$ in supplementary Fig.A5-a. The curves show the expected behaviour where $\bar{J}_0$ increases as $\lambda$ and $J_{reg}$ decreases. In the absence of model errors and for uncertainties accurately



estimated and normally distributed, we should expect $\bar{J}_0$ to be close to 1. Here, for the smallest regularisation we obtain a minimum $\bar{J}_0 \sim 0.6$. For the following seasonal investigations, we choose the value $\lambda = 2500$ which gives a relative error just above one and is located in the area of highest curvature. Additionally, we observe that the results are weakly sensitive to the exact choice of $\lambda$ as the standard deviation of the sliding speed and basal friction computed over all tested values of $\lambda$, except the smallest and the largest one, is generally below 5 %.

**4.1.2   Results**

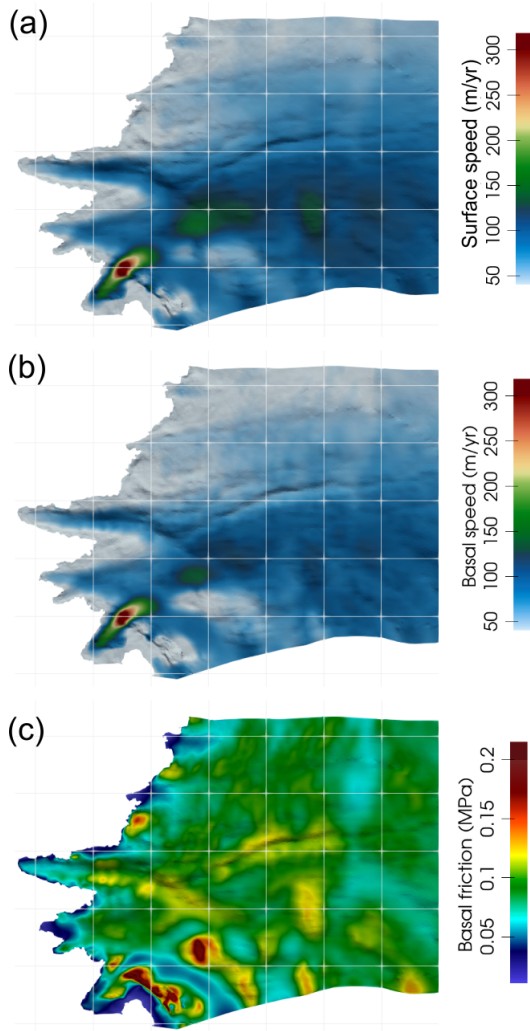

**Figure 3.** Modelled horizontal surface speed (a), basal sliding speed (b) and basal friction (c), inferred from the observed mean winter surface speed (MWS), with basal topography hill-shade added.





The modeled surface speed, sliding speed and basal friction for the mean winter state are shown in Fig.3. The surface velocity shows a good agreement with the observations, with a root-mean-square error of about 3 m/yr, similar to the reported uncertainties ( Fig.A5-b). We notice however for a few places the larger difference (up to 30 m/yr) between the modeled and observed velocity. This is especially true near Ørkendalen Gletscher where some nunataks are not correctly represented in

BedMachine. A similar issue with under-resolved subglacial features may also explain the speed mismatch on the other side of the Ørkendalen valley where BedMachine shows a large uncertainty (>140 m). These areas appear with a particularly high friction compared to the rest of the domain (>0.2 MPa) and nearly no sliding, meaning that the velocities resulting from vertical shear are already larger than the observations, certainly because of the overestimated ice thicknesses. The areas of the friction close to zero along the margin could be the underestimation induced by BedMachine as well, as here, on the steep slopes of

thin ice, the reported error-to-thickness ratio is larger than 50 % (Fig.A2-a).

The spatial pattern of the basal velocity is very similar to that of the surface. We obtain a sliding ratio $u_b/u_s$ of about 0.9 for most of the domain, consistent with deformation profiles measured in the area (Maier et al., 2019). Differences between $u_b$ and $u_s$, what is deformation-induced speed $u_d$ are generally in the order of 10-15 m/yr and can locally increase up to 30-50 m/yr in locations of high traction.

For most of the domain, the inferred basal friction $\tau_b$ is on the order of 0.1 MPa. These values are consistent with previous inversions in the same area performed by Koziol and Arnold (2017) using winter 2008-2009 velocities. Together with a typical sliding velocity of 100 m/yr, this falls in the velocity–traction relationship derived at the catchment scale by Maier et al. (2021), which was interpreted as indicative of hard-bed conditions. However, Maier et al. (2021) also found in this catchment several patches where the bed is weaker than the average. We also observe a few areas where relatively high sliding speeds

correlate with particularly low friction compared to elsewhere, with values lower than 0.05 MPa. This is particularly true for a large fraction of Ørkendalen Gletscher, but also can be seen at the tong of Russell Gletscher. While for the glacier terminus we cannot rule out an under-estimation of the ice thickness that would be compensated by higher inferred sliding speed, this would also be consistent with a "weak" bed offering less resistance due to the presence of deformable till or substantial cavitation. No in-situ measurements have been done to our knowledge on Ørkendalen Gletscher to confirm the presence of till, but seismic

measurements suggest the presence of subglacial sediment within 13 km of the terminus of Russell Gletscher (Dow et al., 2013), in agreement with the weak bed assumed by us.

There is no such low friction under Insunnguata Sermia, where the flow is well constrained by a subglacial valley for the first 25 km inland from the terminus. This is in agreement with Harper et al. (2017) who found no evidence of soft sediments at the bed of Isunnguata Sermia from their network of 32 boreholes. However, no borehole reached the bottom of the valley, where

the friction is locally the lowest, being higher on the adjacent sides. Higher upstream, the subglacial valleys are not aligned with the ice flowlines and, in general, we have higher friction on the leeward sides of the valleys. The spatial heterogeneity of the basal friction shows that it is possible to reconcile opposite views on the nature of the bed in this sector (Booth et al., 2012; Dow et al., 2013; Kulessa et al., 2017; Harper et al., 2017), as over relatively short distances the basal conditions can change from hard to weak bed. Additionally, this could suggest that some specific conditions are required for till accumulation, for

instance topographic depressions or lack of drainage efficiency, forming in the long run the non-uniform bed properties.





## 4.2 Seasonal inversions

### 4.2.1 Model set-up

As the observation errors are not uniform in time, to assess the sensitivity of the summer results to the regularisation, we run
a new L-Curve analysis with the early-July velocity data set. The results are shown in supplementary Fig.A5-a. The minimal
value of $\bar{J}_0$ is now close to 2. This could be due either to an under-estimation of the uncertainties on observed ice speed for
these two weeks, or to the model not being able to exactly match the summer observations. The latter could be due to model
errors not taken into account or because reducing the observations to a standard year leads to more inconsistencies in summer
as there is more variability during this period compared to winter. In addition, we found that the results are more sensitive to
the choice of $\lambda$ than for the MWS inversion, as the standard deviation on $u_b$ and $\tau_b$ between the different tested $\lambda$, except the
smallest and the biggest one, is about 15 %. However, we remark that the dependency of $J_{reg}$ on $\lambda$ is relatively similar to those
obtained with the MWS data sets and the value $\lambda = 2500$ seems consequently also to be a good compromise for this early-July
data set.

To reduce the computational burden for the monthly inversions, we restart from the optimal solution obtained with the
MWS data set and perform a new inversion for each of the 24 time steps of the ice velocity data using the same regularisation
parameter $\lambda$. As we cannot guarantee in general that the given $\lambda$ will be optimal for all the data sets, we stop the minimisation
after 30 evaluations of the cost function to avoid overfitting. This choice is motivated by the fact that in general the cost
function decreases rapidly during the first iterations and then stagnates at a value close to the noise level where it may overfit
the observations if regularisation is not used (Arthern and Gudmundsson, 2010; Habermann et al., 2012). This can be clearly
seen in Fig.A5-b for the spring and summer months, where the error decreases during the first 10 to 20 iterations then stagnates.
Note that the error can locally increase between two successive evaluations of the cost function, as the global convergence is
enforced by a line-search phase so that the minimisation routine effectively checks that the cost function decreases globally.
As expected, because the velocities are relatively stable during the winter months and thus close to the MWS, the error is
already nearly very low with the initial guess, and stagnates or eventually slightly increases (while $J_{reg}$ decreases) during the
iterations.

In Fig.A1, we show the misfit maps as the difference between observed and modeled surface ice speeds (only for the early
half of the months). The periods showing the largest misfit correspond to the transition between different surface conditions.
This is especially true in June, July and September, when the surface changes from snow-covered to melting-ice or vice-versa.
Note that the November dataset seems more likely to be corrupted by poor satellite imagery than by the surface conditions.
The uncertainty in the observed velocity maps are indeed the most important for these periods, often exceeding 30 m/yr, which
represents a significant relative uncertainty in the upper part of the domain where speed is below 100 m/yr. However, the
surface velocity stays captured relatively accurately in the first 50-70 kilometres from the margin, and the inversions give a
good match between the adjacent time steps, giving us confidence in the interpretation of the basal fields in these areas. Taking
that into account, as well as change in basal topography uncertainties, we further demonstrate and discuss mainly the results
on downstream half of the modelling domain.





### 4.2.2 Results

In Fig.4, we present the ratio of the inverted basal friction $\tau_b$ and sliding speed $u_b$ for 10 inversions (out of the 24) to their winter mean state. Results from early October to early March are fairly similar, so we show only early January as an example of this period. From late March to late September, the relative changes in $\tau_b$ and $u_b$ are shown every two weeks. Note that the extreme $\tau_b$ variability in late June and early July over the upper shown area is most likely unrealistic and induced by the discussed quality of the input surface velocity fields.

These inversions demonstrate that $u_b$ doubles first along the ice margin and then the acceleration propagates inland until mid-July. Thus, in the first twenty kilometres from the ice margin, the maximum sliding speed is reached by early June, whereas higher up the peak only arrives later in early July. Similarly, the sliding speed first decreases along the ice margin and then the deceleration propagates inland until the end of September. In late September, the velocity is generally lower than the MWS by about 10 to 20 % in the first 10-15 kilometers from the margin, while it is still slightly higher or equal further upstream.

Following changes in sliding speed, the basal friction $\tau_b$ first decreases along the ice margin in late May. Over time, this decrease then propagates higher up inland, mainly in subglacial valleys and depressions. Note that the lowering of $\tau_b$ is usually correlated with the highest increases in sliding speed. However, as the global force balance must be maintained, this decline cannot be widespread. Thus, the stresses are redistributed locally and simultaneously on higher parts of the bed or on the sides of subglacial valleys where the friction rises. The sliding speed here rises as well, but it is less pronounced. Both increases and decreases in $\tau_b$ over the domain are of the order of $\pm 30$ %. By late September, $\tau_b$ returns to its winter state. The exception being Ørkendalen Gletscher where it remains higher and where we also have the most significant decrease in sliding speed in autumn compared to the mean winter value.

### 4.3 Ice deformation versus sliding speed

An interesting question is how the contributions of deformation and sliding to the surface speed change seasonally.

Ice viscosity and surface topography are fixed in inversion set-ups, therefore the temporal variability in the ice deformation rate occurs only due to the change in the inferred basal friction (thus, in the shear stress). We found that in summer, on average across the domain, the magnitude of the deformation speed $u_d$ increases, but the proportion of surface velocity it represents decreases. The mean winter value of $u_d$ across the domain is about 8 m/yr, rising to about 20-30 m/yr in some topographically predefined locations (Fig.A4-a). In early July it rises by +12 % on average (Fig.A4-b). At the same time, the contribution of deformation to glacier surface motion estimated as a fraction $u_d/u_s$ decreases from winter to summer over the majority of the domain (see Fig.A4-c), with average values of 10 % and 8% respectively. That means the sliding velocity represents about 90 % of the surface flow in winter and even more when velocities increase in summer. Thus, the summer acceleration observed on the surface is mainly due to enhanced sliding and not deformation, what can be also seen in Fig.5-a.

Although direct measurements of sliding velocities and ice deformation rates are rare in Greenland, Maier et al. (2019) estimated them within a network of boreholes located approximately 20 km from the ice margin of the Insunnguata Sermia catchment (blue star in Fig.1). These observations were made during the winter season 2015-2016 in boreholes spaced by







**Figure 4.** Seasonal change of basal sliding speed (left subcolumns) and basal friction (right subcolumns) relative to the winter mean state, with basal topography hill-shade added.

about 150 m, and include GPS (providing the surface velocity), temperature and inclinometry (providing the deformation). At this site, the deformation was found to account for only 4 % of the surface velocities and, thus, ice sliding was responsible

for the overwhelming majority of surface velocity during the winter. In addition, their measurements show that the majority of deformation happens in the first 150 m above the bed, and that almost no deformation occurs in the upper 75 % of the ice column.





We compare the measurements made by Maier et al. (2019) with the modelled fields at the same location in Figs. 5a and 5b, in the form of vertical profiles of the horizontal velocity magnitude $|\boldsymbol{u}_H|$ and shearing rates $\mathrm{d}|\boldsymbol{u}_H|/\mathrm{d}z$. The inverted sliding

and surface velocities in winter of about 105 and 115 m/yr are in agreement with the in-situ measurements of 110 and 114 m/yr respectively. It would appear that the model produces a slightly larger deformation speed $u_d$ than observed (10 m/yr versus 4.6 m/yr in the in-situ observations), with excess deformation mostly coming from the upper 75 % of the ice column and thus not expected to vary in time with changes in basal friction $\tau_b$.

During the velocity peak in early July, the deformation speed $u_d$ in our inversions rises up to 15 m/yr and thus increases

relative to winter by about +50 % (maximum $u_d$ is 19 m/yr in early August, or +90 % from winter, Fig.5-a). At the same time, basal sliding speed $u_b$ increases from 105 to 165 m/yr (+57 %), and surface speed $u_s$ from 115 to 180 m/yr (+57 %). Consequently, at this specific point, the relative increase in surface flow velocities from winter to summer is about 8 % due to an increase in the deformation rate and the remaining 92 % corresponds to the accelerated sliding. Note, that the overall contribution of $u_d$ to the surface speed is almost invariable from winter (9%), being slightly lower at the moment of maximum

glacier motion (8%) and slightly higher when deformation of basal layers intensifies (10 %).

It should be also noted that the actual magnitude of modelled deformation and sliding velocity in this analysis depends on the constitutive law used for the ice rheology (here, Glen's flow law, Eq.2). In this study, we assume a viscosity enhancement factor $E=1$ (see Eq.3) to describe the ability of ice to deform. This means we adopt Glen's rheology parameters constrained via laboratory experiments without any modifications. Larger values of $E$ would correspond to simulation of softer ice, while lower

values of $E$ represent stiffer ice, which provides the adjustment of viscosity imposed by laboratory parameters if required. In Fig.5, we test different values of $E$ against in-situ observations from Maier et al. (2019) by comparing the vertical deformation rate obtained for inversions performed with $E=0.5$, 1 and 2.5 with the mean measured winter deformation profile obtained over 9 boreholes.

With $E = 0.5$, thus stiffer ice, the model reproduces the winter deformation rate well over most of the ice column, but

underestimates it for the lower 100 m above the bedrock; there the majority of shearing happens according to observations. In this case, even with enhanced shearing in summer, modelled deformation does not reach the level observed in winter. With $E = 2.5$, the ice deformation is much larger over the entire ice column than observed, suggesting that the modelled ice is too soft. A value of $E = 1$ provides a good compromise where the deformation rate over the winter months near the bed is similar to that of the in-situ measurements and does not deviate significantly at other depths.

## 4.4   Relation between $\tau_b$ and $u_b$

To discuss further the leading processes behind the seasonal variability of basal fields, we take a closer look at three different locations that are representative of the main types of $\tau_b$-versus-$u_b$ behavior obtained across the modelled domain (Fig.6). As the aim will be to interpret the results in terms of friction law, we show the seasonal evolution of $\tau_b$ as a function of $u_b$ (bottom subplots) together with their evolution during the year (top subplots).

In the first case, at point A located at the Insunnguata valley floor about 10 km from the ice margin (see Fig.1), we have a clear correlation between lower friction and higher sliding speed. The friction slightly rises from January to May but there





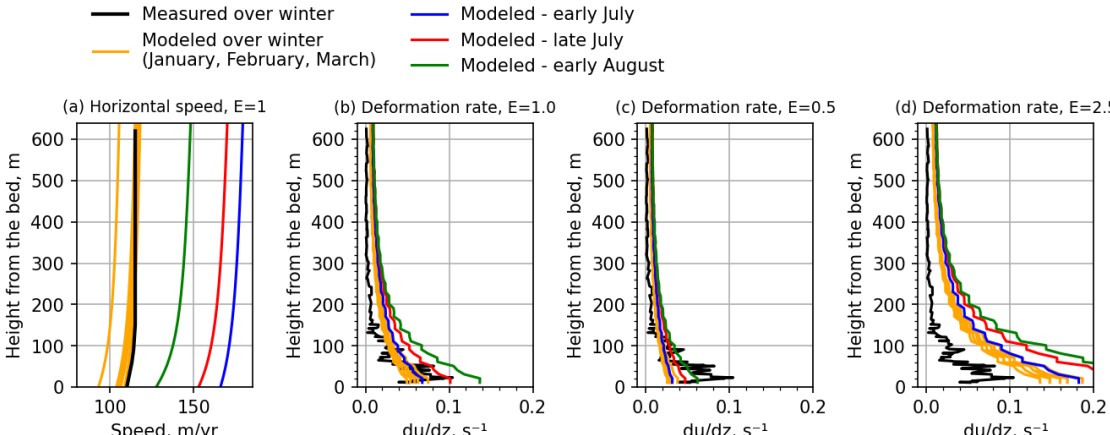

**Figure 5.** Vertical profiles of modelled (a) horizontal velocity magnitude $|\boldsymbol{u}_H|$ and (b)-(d) vertical shear rate $\mathrm{d}|\boldsymbol{u}_H|/\mathrm{d}z$ for varying enhancement factor E at the location of borehole measurements done by Maier et al. (2019), drawn here in black.

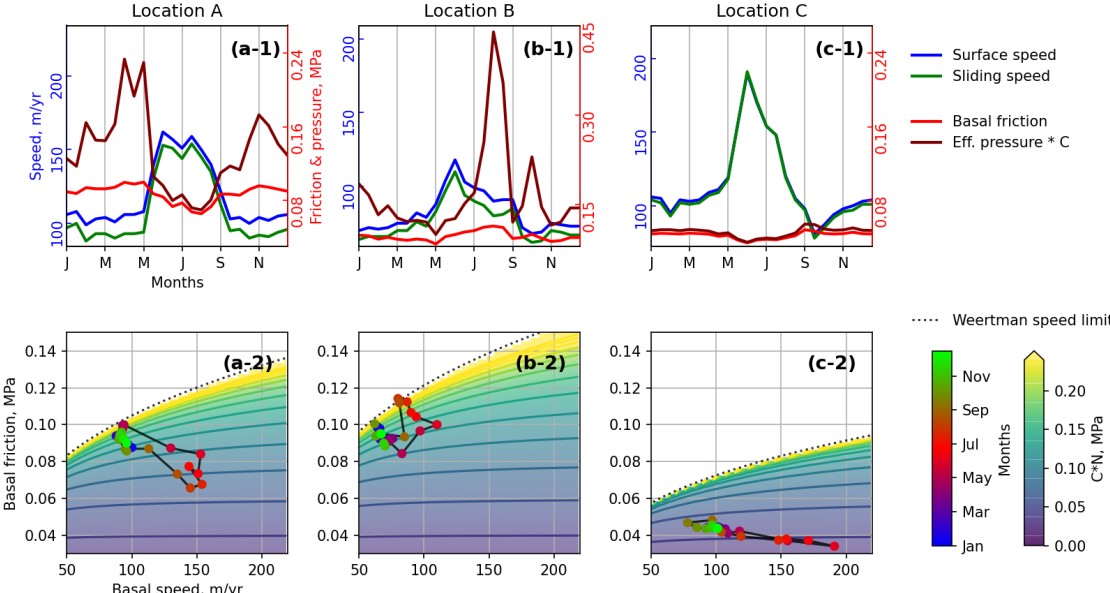

**Figure 6.** Surface speed $u_s$, sliding speed $u_b$, basal drag $\tau_b$ and scaled effective pressure $CN$ at three locations indicated in Fig.1. Top panels show the evolution of $u_s$, $u_b$, $\tau_b$ and $CN$ as a function of time. Bottom panels represent the evolution of the relation between $\tau_b$ and $u_b$ throughout the year, where the colored background shows the scaled effective pressure $CN$ obtained from Eq.12 with the solid lines corresponding to isovalues spaced by 0.02 MPa. The dotted line represents the upper limit corresponding to the Weertman regime (see Sec.5.1).



is no clear trend for the velocity. Notable acceleration starts in early May in conjunction with a decrease in $\tau_b$. High sliding velocities, 65 % above the winter average, are maintained until mid July with some variability. At the same time, the basal friction continuously decreases until August with a minimum at 30 % below the winter average. After that, $u_b$ decreases and $\tau_b$

increases back to winter values. The seasonal evolution of $\tau_b$ as a function of $u_b$ shows a clear hysteresis where: for the same friction, the sliding speed is higher during the acceleration phase (spring) than during the deceleration phase (autumn). The regions that experience this type of behavior are mainly the subglacial valley bottoms where the basal friction is not too low in winter ($\sim 0.1$ MPa).

Point B is positioned on the northern valley slope of Insunnguata Sermia at about the same distance from the ice margin as

point A. Here, during the warm season, the basal friction globally rises ($+20\%$) together with basal sliding ($+80\%$). However, there is a preceding phase from January to May where the sliding speed slightly increases in conjunction with decreasing friction. The clear acceleration in May correlates with an increase in friction. Further, the sliding velocity starts to diminish in June while the friction reaches its maximum only in August. This relation between $\tau_b$ and $u_b$ is generally associated with valley sides and higher basal topography.

Point C is located at Ørkendalen Gletscher about 10 km along the glacier flowline from the ice margin (Fig. 1). This point is typical of areas with lower-than-average friction values ($< 0.05$ MPa), i.e. the base offers little resistance throughout the entire year. Here, the velocity increases almost twofold from late April to late May and is associated with a relatively small absolute change in friction of about 0.01 MPa compared to point A where $\tau_b$ dropped by almost 0.04 MPa. For this point, it is also worth noting that the speed minimum appears in early September, being clearly below the winter mean. As for point A, changes are

mostly anti-correlated, i.e. sliding rises with decreasing friction and vice-versa, and the maximum in $u_b$ corresponds to a minimum in $\tau_b$.

## 5 Basal water pressure

### 5.1 Water-dependent friction law

#### 5.1.1 Definition of the friction law

To interpret the variations of the basal friction and velocity in terms of water pressure, we adopt a friction law that has been proposed originally to represent the flow of clean ice over a rough bedrock with cavitation (Schoof, 2005; Gagliardini et al., 2007). In its simplest form, this friction law relates the basal friction $\tau_b$ to the effective pressure $N$ and the sliding velocity $u_b$ and, following Gagliardini et al. (2007), is expressed as:

$$\tau_b = CN \left( \frac{u_b}{C^n N^n A_s + u_b} \right)^{1/n} \tag{12}$$

where $A_s$ is the sliding parameter without cavity, $C$ is a parameter related to the bed roughness and the exponent $n$ is usually equal to the flow-law exponent in Eq.2. The effective pressure $N$ conceptually represents how frictional forces are reduced by the presence of pressurized water and is equal to the difference between the Cauchy compressive normal stress to the



bed surface and the water pressure $p_w$. Here we approximate the normal stress by the ice pressure $p_i$ solution of the Stokes system, thus $N = p_i - p_w$. In practice, $p_i$ is close to the hydrostatic ice overburden pressure $p_i \approx \rho g H$, and this allows it to be

consistent with observations of the water pressure that are reported as a fraction of the ice overburden pressure. A friction law of a type similar to Eq.12 is used by most ice-flow models coupled with a subglacial hydrology model (Pimentel et al., 2010; Hewitt, 2013; Gagliardini and Werder, 2018), and while it was primarily developed for hard beds, a similar expression has been proposed for deformable beds (Zoet and Iverson, 2020) and has been shown to provide good fit to observations for Pine Island Glacier, Antarctica (Joughin et al., 2019). Some authors (Koziol and Arnold, 2018, 2017; Brinkerhoff et al., 2021) have used a

different friction law where the dependency to $N$ has been introduced in a mostly empirical manner (Budd et al., 1979). While Koziol and Arnold (2018) found that this last law gives a better fit to observations than Eq.12 in a coupled ice-flow-hydrology model of Russel glacier, we do not consider this law as it has less physical background and does not satisfy Iken's bound (Iken, 1981).

From Eq.12 it can be shown that the product $CN$ can be expressed as a function of the fields $\tau_b$ and $u_b$, that are solutions of

the inverse problem, and the parameter $A_s$ as:

$$CN = \tau_b(1 - \tau_b^n u_b^{-1} A_s)^{-1/n} \tag{13}$$

Expressed this way, it is easy to see that this expression exhibits two asymptotic behaviors:

- when $\tau_b^n u_b^{-1} A_s \ll 1$, the relation tends to a Coulomb-type friction law $\tau_b = CN$ and the effective pressure does not depend anymore on the sliding speed $u_b$. That implies a lower bound for Eq.13 $N > 0$, so we cannot get a water pressure

exceeding the ice pressure, even if that is observed with local in-situ measurements (Wright et al., 2016; Hoffman et al., 2016).

- when $\tau_b^n u_b^{-1} A_s \to 1$, $CN \to \infty$, and at the limit the basal friction is described by the classical Weertman friction law $\tau_b = (u_b/A_s)^{1/n}$. Note that in practice $N$ should satisfy the upper bound $N \le p_i$, meaning that the water pressure must remain positive.

Finally, the case $\tau_b^n u_b^{-1} A_s \ge 1$ is incompatible with Eq.13 and means that the inverted values are inconsistent with the choice of $A_s$ as the bed is already too slippery even without water.

### 5.1.2 Choice of the sliding parameter $A_s$

The sliding parameter $A_s$ should depend on the near-basal ice rheology and on the small scale roughness of the bed and thus is likely to be variable in space. Large values of $A_s$ correspond to a slippery bed even without water, while low values represent

a bed that offers more frictional resistance to the flow. Thus, depending on the choice of $A_s$, one can approach the Weertman or Coulomb limits of the friction law defined in Eq.12. This obviously will affect the effective pressure values $N$ retrieved, but also the amplitude of the seasonal changes expected to be observed.

Using a similar method to infer changes in water pressure from several inversions of the basal conditions under Variegated Glacier over a year, Jay-Allemand et al. (2011) compute a spatially varying $A_s$ so that the upper bound $N \le p_i$ is always





fulfilled. In an application to the same area we investigate, Koziol and Arnold (2017) use winter velocities and a modelled winter basal water pressure field to invert for a single parameter in a friction law similar to Eq.12. Their coefficient $\mu_b$ corresponds to our $C$, and they use a constant value for the product $A_s C^n$ (i.e. their $\lambda_b A_b$), so that for us this would correspond to spatially varying $A_s$ and $C$. They found that the whole domain is close to the Weertman regime in winter.

Here, to spatially constrain $A_s$, we first compute the sliding coefficient $A_s^W$ that would be obtained in the Weertman regime
using $u_b$ and $\tau_b$ from the WMS inversion. Properly saying this would correspond to an "effective" $A_s^W$ as it will reflect the effective winter state of the bed roughness which could include the smoothing effects of potentially existing cavities that are not closed or kept open by basal melting water (Cook et al., 2020).

The distribution of $A_s^W$ inferred at the mesh nodes is shown in Fig.7-a. The median value is $4.04 \times 10^{-21}$ m Pa$^{-3}$ s$^{-1}$, corresponding approximately to a basal traction $\tau_b$ of 0.1 MPa for a sliding speed of about 100 m/yr. The same order of
magnitude is found by looking at the relations between $\tau_b$ and $u_b$ inferred by Maier et al. (2021) at the scale of the GrIS catchments that are identified as being subject to hard-bed physics (see authors' Supplementary materials, Fig.S9 and Table S1). It also has a consistent order of magnitude with the value $1.66 \times 10^{-21}$ used by Hewitt (2013) and Gagliardini and Werder (2018) in synthetic applications developed to represent a typical Greenlandic land-terminating glacier such as those in the Russell sector.

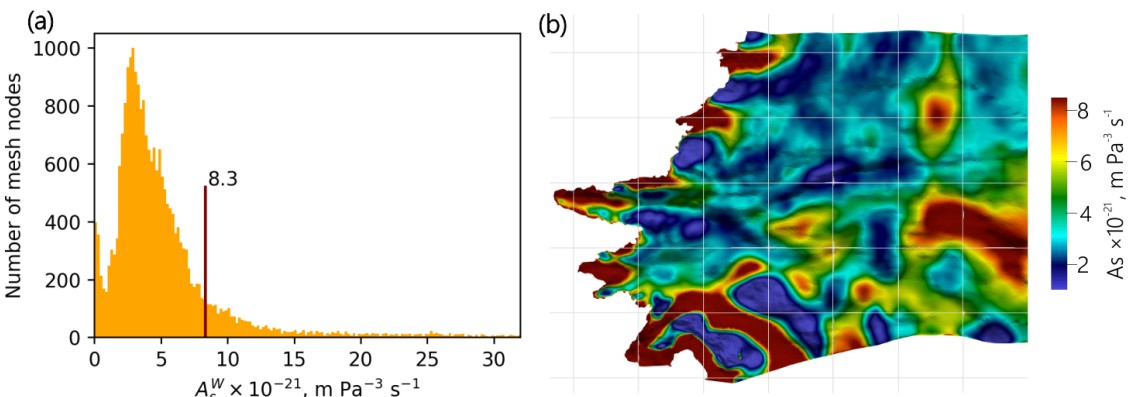

**Figure 7.** (a) Histogram of $A_s^W$. (b) Map of the $A_s$ derived with Eq.15 and used for water pressure calculation, with basal topography hill-shade added.

The spatial distribution of $A_s^W$ mostly reflects the inferred winter basal friction (Fig.3-c), where the areas with a low friction would imply a very slippery bed. As mentioned previously, if we assume that these areas of weak bed might be underlain by deformable till or too many water-filled cavities maintained over the winter, the value of $A_s^W$ has no meaning in its Weertman-law sense and these areas should be described with a Coulomb friction law. It is then reasonable to impose an upper bound for $A_s$. There is no obvious upper limit and we arbitrarily choose $A_s^{MAX} = 8.3 \times 10^{-21}$ m Pa$^{-3}$ s$^{-1}$ (Fig.7-a). That limit covers
85% of the mesh nodes and puts the areas of very-likely assumed weak bed (Ørkendalen Gletscher valley, Russell Gletscher tongue) in the Coulomb regime while leaving the areas of strong bed in the Weertman one.





Directly taking $A_s^W$ to compute $N$ from Eq.13 would certainly lead to incompatibilities for some nodes in some inversions, i.e. $\tau_b^n u_b^{-1} A_s^W \geq 1$. We therefore choose to lower $A_s$ as $A_s^W - 2\delta A_s^W$, where $\delta A_s^W$ is the uncertainties on $A_s^W$ estimated as uncertainty propagation of inferred $u_b$ and $\tau_b$. It is computed as:

$$\frac{\delta A_s^W}{A_s^W} = \frac{\delta u_b^W}{u_b^W} + n\frac{\delta \tau_b^W}{\tau_b^W} \tag{14}$$

where $\delta u_b^W$ and $\delta \tau_b^W$ are the standard deviation of the basal speed and friction obtained from the 6 inversions of the winter months from January to March, as we assume no significant changes in basal conditions over this period. Our winter velocity and friction uncertainties are about 4 % and 2 % respectively on median across the domain.

As more than 90 % of nodes have the January to March variability of $\tau_b$ and $u_b$ values under $\pm 2\delta$ from mean winter state, we consider it sufficient to use $-2\delta A_s^W$ to avoid upper bound incompatibility of Eq.13 (see Sec.5.1.1) across the majority of the area. Therefore, $A_s$ is taken as

$$A_s = \min(A_s^W - 2\delta A_s^W, A_s^{MAX}) \tag{15}$$

which brings together the initial slipperiness assumed under Weertman conditions, scaled down with the respect to uncertainties in modelled basal velocity and friction, and the arbitrary prescribed boundary to deal with weak-bed regions. Fig.7-b represents the final field of $A_s$ used further to infer water pressure, with maroon areas corresponding to the weak-bed regions restricted by $A_s^{MAX}$.

Reducing $A_s$ compared to $A_s^W$ is also consistent with observations in boreholes that suggest locations of relatively high water pressures $p_w$, above 80 % of the overburden pressure $p_i$, even in winter (Van De Wal et al., 2015; Wright et al., 2016). The latter means that our core assumption for winter-based $A_s^W$, which is a lack of pressurized water and a corresponding inhibition of sliding, would be not valid everywhere and thus the estimated slipperiness of the bed in "dry" conditions would be higher that it is.

Note, that even with accurately constrained $A_s$ we stay limited to inferring confidently the magnitude of winter pressure with the assumption of winter basal conditions close to the Weertman regime. As in this regime basal friction is weakly dependent on effective pressure, the small variations of inferred $u_b$ or $\tau_b$ would have a large impact on the retrieved water pressure. Simple pressure calculations with Eq.13 using the median value $A_s^W = 4.04 \times 10^{-21}$, and typically in the Russell sector $\tau_b \sim 0.1$ MPa, $u_b \sim 100$ m/a, and a thickness of 1000 m, shows that $p_w$ values over a very large range, including $> 80$ % of $p_i$, can be obtained with variations of $u_b$ and $\tau_b$ of only a few percent which represents the range of uncertainties on $u_b$ and $\tau_b$.

### 5.1.3 Choice of the bed roughness parameter $C$

Once $CN$ has been inferred from the inversions, a value for $C$ has to be prescribed to translate this in terms of basal water pressure. As shown by Schoof (2005), $C$ should be lower than the maximum local positive slope of the bedrock topography at a decimetre to meter scale, so that the ratio $\tau_b/N \leq C$ fulfils Iken's bound (Iken, 1981). As there is no observational or experimental constraints for the value of $C$, most authors use values that are also consistent with the values that have been





inferred to describe the Coulomb behaviour of deformable beds and that range between 0.17 and 0.84 (Iverson et al., 1998; Truffer et al., 2000; Cuffey and Paterson, 2010; Iverson, 2011).

Here we use a constant and uniform value and take $C = 0.16$ as in the synthetic applications for a typical Greenlandic landterminating glacier by Hewitt (2013) and Gagliardini and Werder (2018). Using a value that might be considered as a lower bound will underestimate the water pressures but over-estimate the temporal variations, thus highlighting the areas where the changes are the most pronounced. In the following, the absolute values of $N$ must then be regarded with caution, however the relative variations remain independent of $C$.

**5.2   Seasonal changes in the modeled basal water pressure**

**5.2.1   Pressure fields**

Using the $A_s$ and $C$ parameters discussed previously, the effective pressure obtained from Eq.13 has been derived for the 24 dates and further unwrapped to water pressure $p_w = p_i - N$ and flotation fraction $FF = p_w/p_i$. The flotation fraction maps inferred for early January and early July are shown in Fig.8. They correspond approximately to the months with the lowest and
highest $FF$ on average, respectively.

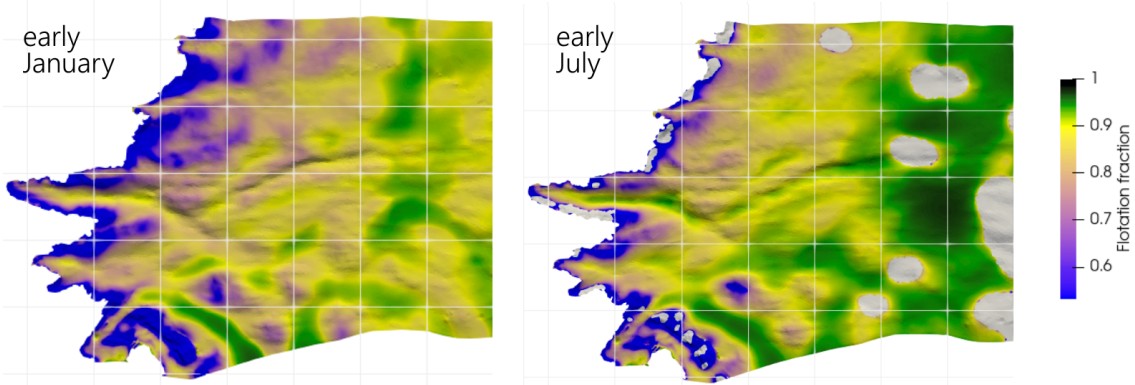

**Figure 8.** Flotation fraction maps obtained for early January (right) and July (left) using a regularized Coulomb friction law from Gagliardini et al. (2007), with basal topography hill-shade added. The grey areas in the right panel have no value as they are outside the validity domain of Eq.13 ($\tau_b^n u_b^{-1} A_s \geq 1$).

Although the absolute pressure values obtained in winter are highly uncertain because we assumed the system to be close to the Weertman regime during this period, we obtain $FF$ values above 0.8 for most of the domain in agreement with measurements obtained in boreholes in the range of 0.8–1.1 (Meierbachtol et al., 2013; Van De Wal et al., 2015; Wright et al., 2016). A good coherence between the variability of the spatio-temporal pressure fields and various constraints, e.g. basal topography
or ice thickness, is successfully obtained as well. Between early January and July (Fig.8), the $FF$ increases globally by 7 % (median $FF$ over the domain 0.83 and 0.88 respectively). The more pronounced increases happen mainly in valley floors and troughs, where the system comes very close to the flotation limit ($FF = 1$) during summer. This would be consistent with the





idea that the water follows hydraulic potential gradients and concentrates in these topographic depressions (Wright et al., 2016; Downs et al., 2018). The weak-bedded areas, which are considered to be in a near-Coulomb regime in winter, have a higher

flotation fraction than surrounding areas over the entire year. This is especially true for Ørkendalen Gletscher, where $FF$ is already above 0.95 in early January. In summer, in those regions, $FF$ cannot increase significantly as from defined boundaries for Eq. 13 $CN$ is always positive, meaning that the water pressure is always lower than the ice overburden pressure. We also observe that $FF$ generally rises with the distance to the ice margin, similarly to the modelling of water routing system steady-stat by Meierbachtol et al. (2013). This could suggest an increased drainage efficiency when approaching the terminus and vice

versa when moving towards the interior of the ice sheet.

Modelling of the water pressure using subglacial hydrological models (de Fleurian et al., 2016; Koziol and Arnold, 2017; Downs et al., 2018) shows $p_w$ spatial patterns similar to those obtained by us: the $FF$ increases with the distance from the margin and is higher in large troughs in the glacier bed. We note however that such hydrological models generally obtain lower water pressure values than estimated here or observed in boreholes, with a typical range of winter $FF$ values 0.4 to 0.7 across

the Russell sector. In de Fleurian et al. (2016), the modeled $FF$ increases significantly in summer compared to the winter mean for altitudes above 1000 m (about 45 km inland from Insunnguata front line), while below it changes are moderate and mainly concentrated in the Insunnguata valley. While it is difficult to conclude from our inversions on the pressure changes above 1000 m, it seems that the pressure variations below it are more pronounced and systematically higher than modelled by de Fleurian et al. (2016) (this result can also be seen in Fig.9 and 10), which might suggest that the subglacial hydrological

system is not as efficient as assumed in this part of the ice sheet. In other similar work, Downs et al. (2018) reproduced even larger $FF$ variability from summer to winter than we do, adjusting the seasonally evolving hydraulic conductivity. However, they still have absolute values of FF in winter almost half of inferred by us.

### 5.2.2 Physical processes driving the seasonal dynamics

Isovalues of $CN$ computed from Eq.13 are reported in Fig.6 (bottom subplots) for the three particular locations (points A, B

and C) discussed previously. In addition, we show in Fig.9 and Fig.10 the seasonal evolution of basal velocity, friction and flotation fraction along two profiles A' and C' that pass through point A and C, respectively (see Fig. 1).

Points A and B, that we have identified as having a hard bed, are, by assumption, close to the Weertman regime in winter and it shows that small variations in either $u_b$ or $\tau_b$ have a large effect on the magnitude of the retrieved effective pressure.

At point A, the relationship between $\tau_b$ and $u_b$ does not follow a power friction law where the evolution of $\tau_b$ would be

proportional to $u_b^{1/n}$ such as in the Weertman law (Weertman, 1957). In summer, point A is in a regime closer to Coulomb where changes in $N$ are mainly driven by changes in $\tau_b$ and are fairly insensitive to variations in $u_b$ (Fig.6). In profile A' (Fig.9), the first 20 kilometers seem to follow the same behavior as point A, suggesting that the valley-bottom behavior is compatible with a response of the system to local variations in water pressure; i.e. the water pressure increases when the hydrological system is not able to drain the incoming amount of water efficiently and decreases later in the season as the system gets more

efficient and/or the flux of water reduces.





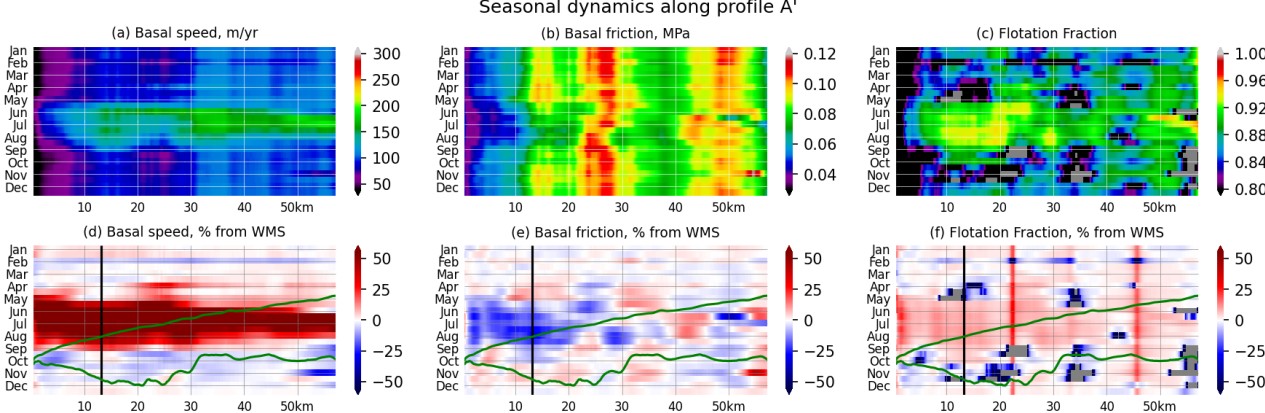

**Figure 9.** Modelled basal speed $u_b$, basal friction $\tau_b$, and flotation fraction $FF$ along profile A' (see Fig.1). (a) to (c) - absolute units, (d) to (f) - fraction relative to the mean winter values (average of January, February, March). In (d) to (f) subplots the vertical black line represents the location of Point A plotted in Fig.6a (see Fig.1), and the green lines represent the glacier top and bottom surfaces with 5x vertical scale factor. The dark-gray areas on (c) and (f) panels have no value as they are outside the validity domain of Eq.13 ($\tau_b^n u_b^{-1} A_s \geq 1$).

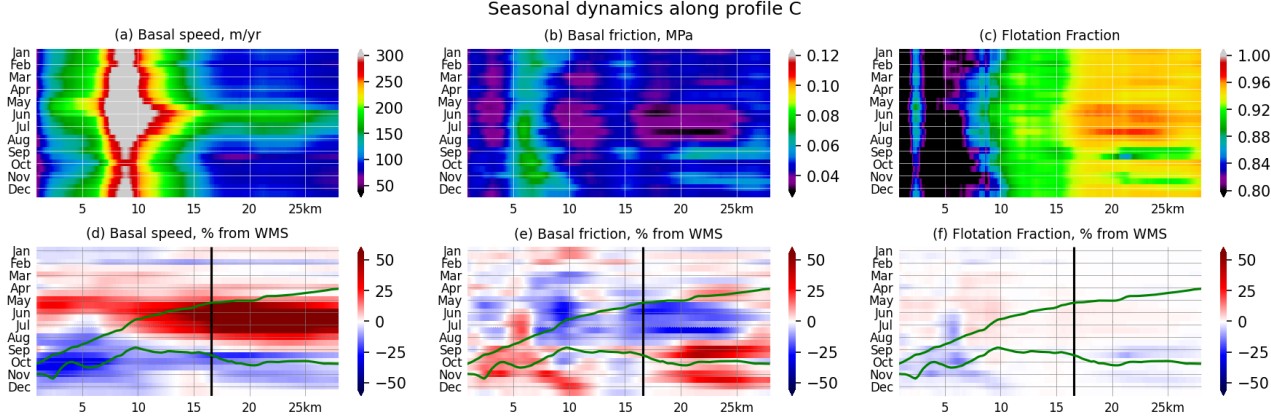

**Figure 10.** Modelled basal speed $u_b$, basal friction $\tau_b$, and flotation fraction $FF$ along the profile C' (see Fig.1). (a) to (c) - absolute units, (d) to (f) - fraction relative to the mean winter values (average of January, February, March). In (d) to (f) subplots the vertical black line represents the location of Point C plotted in Fig.6-c (see Fig.1), and the green lines represent the glacier top and bottom surfaces with 5x vertical scale factor.

In contrast to point A, point B follows a power friction law. The variations inferred at point B are relatively compatible with the Weertman regime for the whole year, so it is difficult to discuss with confidence the temporal variations of $N$ (including sharp picks of $CN$ in Fig.6-b-1). The most plausible hypothesis is that changes in point-B-like areas are mainly driven by changes in longitudinal stresses and are not a response to local variations in water pressure. As the global forces balance that





retain glacier on place should be maintained in summer, while some locations show drop of friction (like valley-bottom point A does) it should be adjusted somewhere else (for instance, on the valley side, in point B). That could explain why the friction peak is observed in August at point B when it is the moment of lowest friction at point A. As such, the behaviors at point A and B could also be respectively related to the active and passive regions as discussed by **?**. As for the slight increase in velocity from January to March correlated with a decrease in $\tau_b$, it would be consistent with a weakly connected hydrological system

that slowly recharges in winter by the basal melting water (Hoffman et al., 2016; Cook et al., 2020).

    Beyond 30 km inland, profile A' (Fig.9) does not follow the bed trough from Insunnguata Sermia and covers higher basal topography where the basal drag alternates between reduced and increased summer values. Along this rugged terrain, the inverted behavior is probably a mix between point A with local variations of the water pressure and point B adjusting to non-local longitudinal stresses.

Point C is located in a region previously assumed as having a weak bed (See Sec.5.1.2). The 30km profile C' (Fig. 9) is mainly goes along the same conditions. It follows a topographical valley where the basal friction is very low (<0.05 MPa), the exception being the ridge located at 3 km where basal drag is more important (∼0.07 MPa). As $\tau_b$ and $u_b$ are weakly connected to each other in this case, we probably approach a Coulomb-like behavior where $\tau_b$ would be proportional to the effective pressure during the whole year. In the hypothesis of the hard bed physics used here, this could be consistent with important

cavitations and therefore low effective pressure throughout the year. Nevertheless, as some reasons should exist to keep the big and numerous cavities open even in winter while the sliding speed here is relatively moderate, we are more inclined to assume the local presence of deforming sediments below the glacier, like those observed in-situ by Dow et al. (2013) for another location on Russell Gletscher, which shows a similar $\tau_b$-vs-$u_b$ relationship in our inversions. The exception to this in profile C' is the aforementioned ridge at 3 km, which appears to offer enhanced flow resistance, similar to point B where we have

posited the existence of hard-bed physics adjusting to longitudinal stresses. As expected for weak beds, changes in effective pressure are mainly proportional to changes in $\tau_b$ and are weakly sensitive to $u_b$. We observe anti-correlated variations between $u_b$ and $\tau_b$ with a maximum in velocity and water pressure in June and a minimum in September; during the winter months, water pressure and velocities increase slowly. As most of this valley is close to flotation throughout the year, the moderate relative increase in water pressure has a large impact on glacier flow in summer as would be expected for a bed described by

a Coulomb friction law. These results are compatible with the model of Bougamont et al. (2014) for weak beds, where the water volume and pressure in the sediments increase in the early melt season leading to a reduction of their shear strength. In late summer and fall, grain re-arrangement leads to increased porosity, reducing the water pressure and higher inter-grain contact, resulting in an overall strengthening of the sediment relative to its pre-summer state, and thus a maximum friction and a minimum sliding speed in fall. It is interesting to note that the decrease in sliding velocity at the end of the melt season

(August to October depending on the surface altitude) appears on general to be much more pronounced in areas where the bed is described as weak than in the rest of the domain where basal conditions appear to correspond to a hard bed.





### 5.2.3 Comparison with runoff: timing and maximum values

Water pressure variations in the system are globally controlled by the increase of surface runoff that percolates to the glacier bed continuously through moulins or crevasses, or sporadically by the drainage of supraglacial lakes (Smith et al., 2015; Stevens
et al., 2016). These water pressure variations are therefore closely related to runoff and drainage-system evolution. Thus, we compare the seasonal evolution of the runoff obtained from the Regional Atmospheric Model (MAR, https://mar.cnrs.fr/) with the inferred effective pressure variations and ice-motion acceleration across different altitudes of our model domain (Fig.11). The runoff is obtained from MAR v3.1, forced by the climate reanalysis ERA5, on a daily basis with a 15 km grid resolution downscaled to 1 km with respect to the surface topography (Fettweis et al., 2020). Further, the period from 2015 to 2019 is
averaged in the same way as for the flow velocities (see Sec.2).



**Figure 11.** Evolution in time and space (averaged over 2 weeks and 20 m elevation bins) of MAR-modelled surface runoff (a), effective pressure $N$ (b), and modelled surface speed $u_s$ (c). The black, dark grey, light grey dots represent the onset, maximum and end of the melting season in (b) and (c). The size of the dots is proportional to the number of mesh nodes found for each state. We define the melting period as runoff $>10$ mmWe m$^{-2}$ week$^{-1}$.

As the domain mostly appears to obey hard-bed physics, the variations in Fig.11 are most probably mainly characteristic of this type of bed. For all altitudes, the water pressure peaks about 2-4 weeks after the start of the melt season as indicated by the MAR runoff. This delay may be due to water percolation duration (Fountain and Walder, 1998), a sliding activation threshold that is not yet reached (Davison et al., 2019), or simply to the fact that the effective pressure changes are still too
small to be observed in the 2-week-averaged surface velocities. Ice-flow velocity begins to increase at about the same time as the effective pressure decreases. Ice speed quickly reaches its maximum a few weeks later, while runoff continues to increase and is maximal in late July at all altitudes. This probably illustrates the fact that since some moment the hydrological system is efficiently able to evacuate the additional meltwater inflow, which does not allow the effective pressure and thus sliding velocity to evolve any further. Once the melt season is over and runoff tends toward zero, the effective pressure quickly returns to its
winter state and does not change significantly until next spring. Up to about 1000 m in surface elevation, ice flow experiences a minimum speed that always occurs after the end of the melting season around late September or early October. This seems to correspond to a minimum in water pressure, but, as we are getting closer to the Weertman regime, the pressure values obtained from the inverted basal friction are uncertain.





The overall evolution over the complete year of the surface speed and the water pressure obtained from the inversions corresponds well with the behaviour expected from previous observations and the current understanding of the interaction between the subglacial hydrological system and the ice flow (Nienow et al., 2017; Davison et al., 2019). For instance, the synchronous increase in basal water pressure and in surface velocity demonstrated here at the beginning of the melt period fits well with the in-situ observations (Bartholomew et al., 2010; Sole et al., 2013; Van De Wal et al., 2015). In our inferred fields the maximum water pressure and velocity are reached while surface runoff still continues to increase, which also corresponds to the local (Bartholomew et al., 2010; Sole et al., 2013) and larger-scale (Sundal et al., 2011; Fitzpatrick et al., 2013) observations in this region, suggesting that a more efficient drainage system is limiting the increase in water pressure at the glacier bed. The drop in water pressure after the end of the melt season (Fig.11-b) corresponds to the presence of a more efficient drainage system established during summer that allows for easy evacuation of the remaining water inputs, chiefly the production of water at the bed as surface melting ceases. Finally, the water pressure slowly increases over winter as the hydrological system becomes inefficient at draining water. Both stages of this evolution have been observed with boreholes measurements as well (Van De Wal et al., 2015).

Thus, we can conclude that it is possible to obtain robust information on the seasonal evolution of friction and sliding at the base of glaciers by using inverse methods on dense time series of surface flow velocities. Using a pressure-dependent friction law in a suitable and well-restricted framework, it is also realistic to relate these changes in basal conditions to the evolution of the hydrological system and in particular the water pressure at the base.

## 5.3 Discussion

In summary, spatially extended time series of ice velocity that can now be obtained at a much higher frequency allow the monitoring of glacier motion at a seasonal scale and thereby offer the opportunity to explore in much more detail the glacier physics driving these changes. Inversions of this time series in the Russel sector indicate that the basal friction changes are mostly consistent with hard-bed physics, which has implications for the choice of friction law in ice-flow models and the development of subglacial hydrologic models. In some small areas, the relation between friction and basal sliding suggests rather the presence of weak beds. In such cases we are not able to confidently conclude from our inversions if this indicates the presence of deformable till, explaining the low strength of the bed, as the results are still compatible with a hard bed with sustained and substantial cavitation.

To relate basal friction and sliding to water pressure, we use a regularised Coulomb friction law designed for a hard bed with cavitation, which seems to give realistic results despite an approximate knowledge of some of the flow-law parameters. The results highlight the presence of active (point A) and passive (point B) locations in terms of hydrology. This is similar to the in-situ observations made by ?Ryser et al. (2014) and Young et al. (2019), with the difference that the inversions offer new insight into the large-scale spatial distribution of active/passive sectors over an extended area. Thus, it is possible from such inversions to infer which regions are hydrologically forced during the melt season and relate it to the spatial evolution of the drainage system. It appears that in the Russell area, the main subglacial water pathways are located in the topographic troughs





of the bed and thus correspond mainly to the active regions, while the "ridges" would correspond to the passive regions, thus providing additional resistance to enhanced flow in the active regions.

Regardless of pressure assumptions, we show here that inversions of basal conditions from time series of seasonal flow-velocity observations provide valuable information on sliding velocity and friction. These inversions could be used to better constrain the results obtained from subglacial hydrology models, as well as to couple these models with ice-flow models.

We note however that efforts would still be needed to obtain sufficiently accurate observational data to constrain seasonal variations in friction and sliding above the ELA (about 50-60 km from the ice margin in this sector of the ice sheet).

In addition, even though the spatial and temporal resolution of the remotely sensed observations seem to be suitable for describing the weekly-to-monthly evolution of glaciers, it is important to note that the interactions between flow dynamics and subglacial hydrology still occur at much higher frequencies (hourly to daily) that remotely-sensed observations are currently unable to capture; a situation that seems unlikely to change in the coming years. On a more positive note, it nonetheless appears that the time-integrated results we obtain are coherent despite the fact that the effects of the variations are indeed at higher frequencies.

## 6 Conclusions

In this paper, we explore the ability of an existing inverse method to use satellite-derived seasonal velocity maps to infer seasonal variations in basal conditions. Based on the observations from multiple satellite missions, we reconstruct the bi-weekly seasonal evolution of surface velocity in a land-terminating sector of the ice sheet in southwest Greenland. Then, we invert 24 velocity fields to obtain the corresponding evolution of sliding speed and basal friction during a typical year, and use them to infer the water pressure changes using a pressure-dependent friction law.

The uncertainties in the inverted fields appear small compared to the amplitude of the observed variations, which allows the seasonal evolution to be reconstructed and the results to be interpreted in terms of water pressure variations. It seems from the winter inversions that this region of Greenland is globally dominated by a sliding regime described by the physics of hard-rock beds and similarly behaving non-deforming till, with small areas showing characteristics corresponding to beds with Coulomb-type sliding, which can be related to the presence of soft sediments or substantial year-round cavitation over a hard bed. This finding differs from previous results that relied either only on hard-bed or weak-bed regimes for this region and, as a consequence, concluded that either a general power law of the form $\tau_b = C_n N u_b^n$ (Koziol and Arnold, 2018) or a pure Coulomb friction law of the form $\tau_b = CN$ (Bougamont et al., 2014) was best suited to couple subglacial hydrology and ice-flow models. Indeed, with the proper set of coefficients, regularized "Coulomb" friction provides a complete description of both hard and weak regimes of the bed physics.

The obtained water pressure variations are consistent with those expected for both the Weertman-like and Coulomb-like bed, and seem to be in phase with the independently derived runoff variations. Thus, we show that inversions of observed surface velocities could serve as an intermediate validation for subglacial hydrology models, assuming that the errors in observed ice dynamics and geometry are small enough to obtain robust inversions. Moreover, the current and future development of space



missions suggests that we will be able to perform the observations with sufficient spatial and temporal resolution to describe the weekly-to-monthly evolution of glacier dynamics on a large scale.

*Data availability.* All datasets used here are publicly available. Elmer/Ice code is available here: http://elmerice.elmerfem.org/. BedMachine data are available from (Morlighem et al., 2017). Note: velocity time series and results from the inversions will be published after revision on an online archive such as Zenodo.

*Author contributions.* A.D. processed the velocity data, performed the analysis, wrote the article, and prepared the figures. J.M. provided supervision, performed the analysis, and wrote the article. F.G.-C. provided supervision, designed and performed the ELMER-related processing, and wrote the article. N.M. processed the in situ deformation data. S.C. proof read the article. All co-authors helped with discussing and reviewing the article.

*Competing interests.* The authors declare no conflict of interest.

*Acknowledgements.* Elmer/Ice computations and ice velocity time series presented in this paper were performed using the GRICAD infrastructure , which is supported by Grenoble research communities. J.M. acknowledges support from the French National Research Agency (ANR) grant (ANR-19-CE01-0011-01). We thank Neil Humphrey for the inclinometery data funded by the NSF Office of Polar Programs-Arctic Natural Sciences awards #1203451 and #0909495, SKB, NWMO, Posiva Oy, and NAGRA.


## Appendix A: Supplementary material

**Figure A1.** Mismatch between modelled and observed velocities (top subplots per section) and error on observed velocities (bottom subplot per section) for the first half of each month.





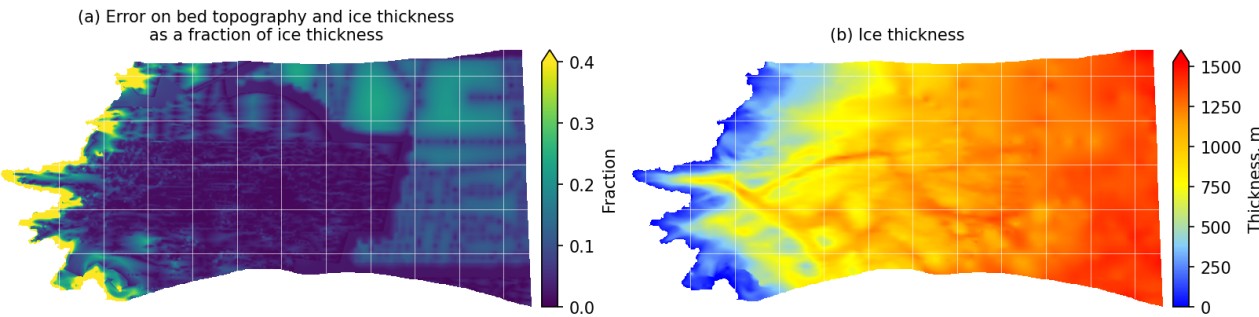

**Figure A2.** The data provided by BedMachine v3 (Morlighem et al., 2017): (a) errors on ice thickness and thus bed topography; (b) ice thickness

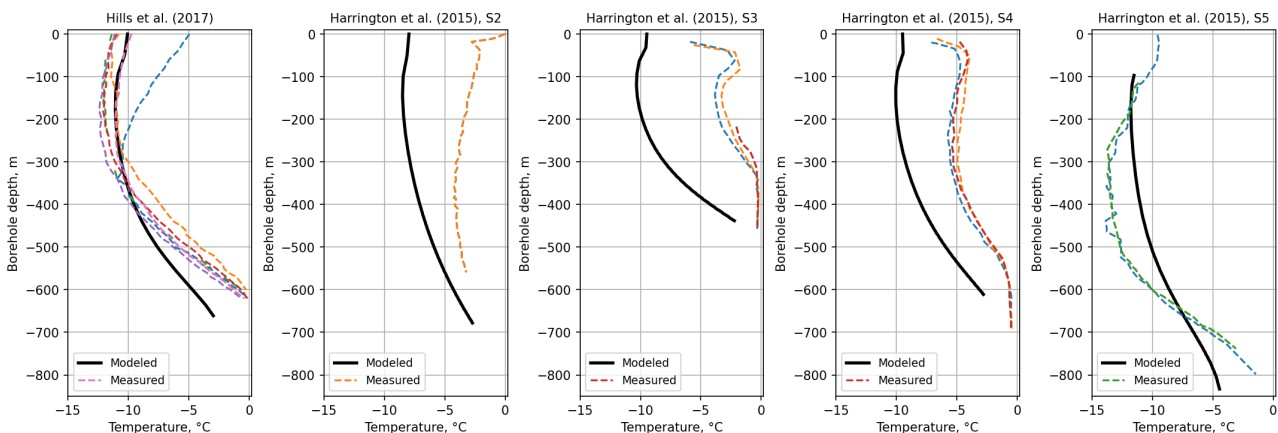

**Figure A3.** Comparison of ice temperature measured in boreholes (Hills et al., 2017; Harrington et al., 2015) with that modelled in SICOPOLIS (Goelzer et al., 2020).

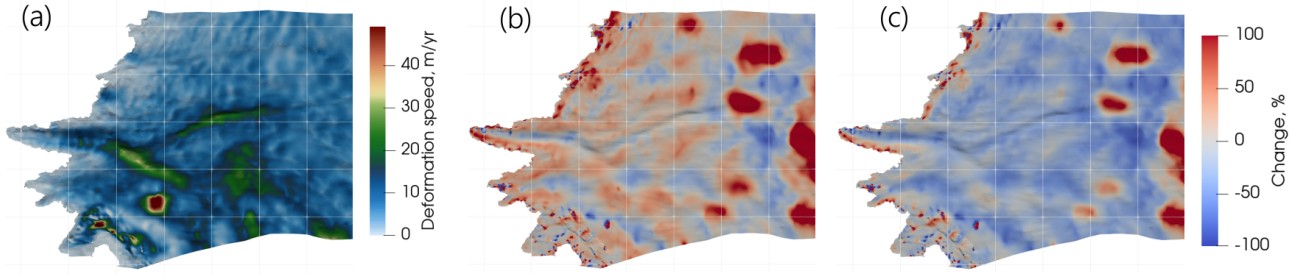

**Figure A4.** (a) Magnitude of modelled deformation velocity $u_d = u_s - u_b$ in early January. (b) Change in deformation velocity magnitude from early January to early July. (c) Change in fraction given by deformation velocity in total ice surface motion $Frac = u_d/u_s$ from early January to early July. All maps are with basal topography hill-shad added.





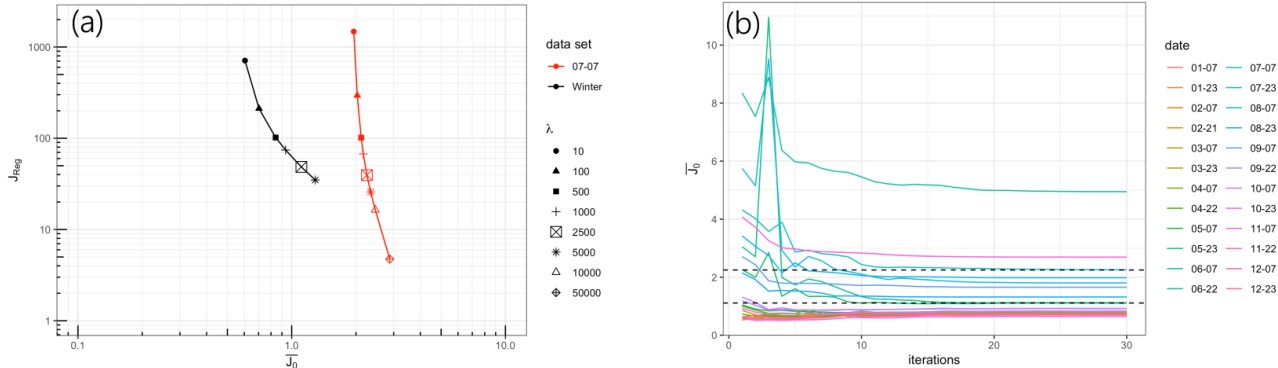

**Figure A5.** (a) Mismatch between measured and modelled surface speed for different regularization parameters $\lambda$. (b) The change in mismatch between measured and modelled surface speed depending on the number of model regularization iterations per bi-month dataset.



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
