# Peer review of "Seasonal evolution of basal environment conditions of Russell sector, West Greenland, inverted from satellite observation of surface flow"

_The Cryosphere, 2021_

## Community Comment (CC1)

First off, I found this an interesting paper.

I want to write a brief comment about the conclusions of the authors, and in particular, how the authors relate their work to Koziol and Arnold (2018) [which I'll refer to as KA2018]. Note, I am the Koziol part of that publication.

Lines 746-749 of the conclusions present a contrast between KA2018 and Bougamont et al (2014), where the former is said to conclude a general power sliding law is best suited while the latter is stated as concluding a Coulomb Law is best suited. The authors then present their work as showing that a regularized Coulomb Law as providing a complete description encompassing both.

This framing misses important context in KA2018. KA2018 never make an unequivocal assertion that the general power sliding law is 'best suited'. Indeed we state that w.r.t the to the Schoof/Gagliardini sliding law there is "… comparable fit to measured velocities for large segments of the velocity time series", as well as that the generalized power law has practical value but "the form and parameters of the sliding law remain uncertain". The full quotations are at the end of this document.

KA2018 avoided the conclusion that one sliding law was 'best' for a few reasons. The first was that the aim of our study was not to investigate which sliding law is correct, but to gain insight about future ice sheet velocities in the context of increased melting. Secondly, we don't have strong evidence from the predicted time series against either of the sliding laws. Here it's important to keep in mind that discrepancies between the recorded and predicted time series can arise from a multitude of data and modelling considerations (e.g. the accuracy of the melt volume and timing, the mathematical formulation of the subglacial hydrological components, the lack of an elastic ice sheet response, the parameters used, …).

I'd like to encourage the authors to reconsider their conclusions in this light, as I don't think KA2018 supports the clear assertion the authors make that KA2018 conclude that power law form is 'best suited'; further, although KA2018 didn't proceed with Schoof/Gagliardini sliding law, I believe our outputs supports its use in the region.

Minor Comments
Line 495-499 should be amended to reflect the fact that KA2018 used both a Schoof/Gagliardini sliding law (i.e. your equation 12) and Budd sliding law, not exclusively the latter as it presently reads.

The two quotes from KA2018 referenced above:

Section 3.2 "The Schoof and Budd sliding laws result in model output of comparable fit to the measured velocities for large segments of the velocity time series. However, during periods of high velocities, the Schoof law can overpredict the magnitude of the velocity by a factor of 3. Model output with the Schoof sliding law is also observed to have a sharper and higher magnitude summer speedup, as well as a slight increase in velocity variability. Since the Budd sliding law results in an overall better match to the measured velocities, the analysis of the velocity time series in the remainder of the paper focuses on those results.

Section 4.2: "Simulation results show the Budd sliding law with standard exponent values has practical value in simulations. However, the form and parameters of the sliding law remain uncertain, and the Schoof law has greater theoretical support (Hewitt, 2013)."

---

## Author Response (AR1)

**Reviewer #1**

- l. 9 - are these modelled or observed water pressure variations? Might be worth specifying here.

**Answer**:

These are modelled water pressure variations. We will update the text accordingly.

**Correction**:

Rewritten  (line 9 in a revisited text)

- l. 84 - "observed geometry" - the problem is, most velocity observations aren't contemporaneous with surface observations - it would be good to show that the impact of this is low.

**A:**

As discussed in lines 179-181, we consider the cumulative surface elevation changes over about 20 years (lag between elevation and velocity data collection) here to have a minor impact on our inversion (e.g. driving stress). Based on the cited study of Helm et al. (2014), the average thinning rate in 2011-2014 was about -1 m/yr. Similar values were also found by Csatho et al. (2014) for the years  2007-2008 and 2008-2009, and by Yang et al. (2019) for the period between 2002 and 2012 (about -0.6 m/yr). The total estimated thinning of roughly 20m is much smaller than the average ice thickness over the area.

In terms of seasonal fluctuations studied here, we do not expect that this will give a major change in surface elevation either. In-situ GPS measurements made at several locations in the region show only minor seasonal surface changes on the order of one meter (Bartholomew et al. 2010, 2011; Cowton et al. 2016; Nathan Maier's personal communication ). We give the detailed rates below for the l. 180-184 comment.

Therefore, while it is possible that using a non-contemporaneous surface with velocity observations induces biases in the inversions, these biases should remain negligible compared to other sources of uncertainty (e.g. errors on velocity, ice thickness or rheology parametrization) and staying almost identical over seasonas should not influence the inferred results of seasonal evolution of the basal conditions.

*Helm, V., Humbert, A. and Miller, H. (2014) 'Elevation and elevation change of Greenland and Antarctica derived from CryoSat-2', Cryosphere, 8(4), pp. 1539–1559.*

*Csatho, B. M. et al. (2014) 'Laser altimetry reveals complex pattern of Greenland Ice Sheet dynamics', Proceedings of the National Academy of Sciences of the United States of America, 111(52), pp. 18478–18483.*

*Yang, Y. et al. (2019) 'Space-Time Evolution of Greenland Ice Sheet Elevation and Mass From Envisat and GRACE Data', Journal of Geophysical Research: Earth Surface, 124(8), pp. 2079–2100.*

*Bartholomew, I. et al. (2010) 'Seasonal evolution of subglacial drainage and acceleration in a Greenland outlet glacier', Nature Geoscience. Nature Publishing Group, 3(6), pp. 408–411.*

*Bartholomew, I. et al. (2011) 'Supraglacial forcing of subglacial drainage in the ablation zone of the Greenland ice sheet', Geophysical Research Letters, 38(8), pp. 1–5.*

*Cowton, T. et al. (2016) 'Controls on the transport of oceanic heat to Kangerdlugssuaq Glacier, East Greenland', Journal of Glaciology, 62(236), pp. 1167–1180.*

**C:**

Rewrited (lines 187-190)

- l. 122 - suggest alternative terms to "master/slave - see, e.g., https://comet.nerc.ac.uk/about-comet/insar-terminology/"

A:

We will replace the terms master/slave by "primary" and "secondary".

C:

Rewritten (lines 123-124)

- l. 132 - vx/vy or vx/vy (l. 157)?

A:

We mean vx/vy. This will be corrected.

C:

Corrected (line 124)

- l. 138 - presumably the MWS map is the median of January/February/March?

A:

No, MWS refers to the mean of these three months. We use it instead of the median because the variability between months is so small that the mean and median are almost identical. We will clarify this in the manuscript.

C:

Rewritten (line 141)

- l. 141 - what are typical values of n here?

A:

n represents the number of speed measurements at each pixel and is highly variable in space (see Fig.2-e) and time. Close to the ice front, the range of n is between < 5 images in winter inland to > 60 in summer. A sentence will be added to better describe the typical range of n.

C:

Rewritten (lines 145-147)

- l. 161-167 - this seems like a reasonable (and interesting!) explanation, but wouldn't these changes also have an impact on the magnitude of the velocity vector?

A:

Yes, the effect would also have an impact on the magnitude, but we estimated that on average across the year this error is smaller than errors from other sources (e.g. related to the geometrical resolution of source imagery). For the most extreme cases (the lowest sun angle in spring/autumn and when only optical imagery is used), we theoretically estimate that the magnitude will be overestimated by less than 10% compared to the real speed. That would correspond to a bias of less than 10 m/yr for the typical speed in this sector of 100 m/yr. As the issue affects only a few time-steps of the velocity database and the bias is compatible with average uncertainty for those months, we did not apply any special corrections on magnitude. This explanation will be more clearly described in the revised manuscript.

C:

Rewritten (lines 171-173)

- l. 180-184 - it would be good to back this up using citations/example data, if possible. Are there any GPS observations for the area that demonstrate this (e.g., from Maier et al. 2019)?

A:

Please see the response to the first comment for the multi-annual surface elevation changes that have been observed. We will complement the references accordingly.

Seasonal rates of surface elevation change have been recorded at several GPS stations installed in the region along a flowline (Bartholomew, 2010, 2011, Cowton, 2016). These observations show small seasonal ice surface changes of less than 1.5m, and more commonly this change is less than 0.5 m. Further, they are interpreted by authors to be partly induced by glacier uplift, thereby the ice thickness changes are even smaller. This is consistent with the seasonal rates recorded from GPS stations in 2014-2017 used in Maier et al. 2019 which showed seasonal uplift of about 0.25 m/yr (currently unpublished; personal communication). We will add the values for the rate and citation in the text.

*Bartholomew, I. et al. (2010) 'Seasonal evolution of subglacial drainage and acceleration in a Greenland outlet glacier', Nature Geoscience. Nature Publishing Group, 3(6), pp. 408–411.*
*Bartholomew, I. et al. (2011) 'Supraglacial forcing of subglacial drainage in the ablation zone of the Greenland ice sheet', Geophysical Research Letters, 38(8), pp. 1–5.*
*Cowton, T. et al. (2016) 'Controls on the transport of oceanic heat to Kangerdlugssuaq Glacier, East Greenland', Journal of Glaciology, 62(236), pp. 1167–1180.*
*Maier, N. et al. (2019) 'Sliding dominates slow-flowing margin regions, Greenland Ice Sheet', Science advances. American Association for the Advancement of Science, 5(7), p. eaaw5406.*

C:

Rewritten  (lines 187-190)

- l. 201 - why 0.9 m?

A:

This value is not physically meaningful; it merely allows an easy separation between ice-covered and ice-free areas in the model. The model does not support meshes that have null thickness. In order to include ice-free areas, we therefore impose the arbitrary value of 0.9 m thickness for them. This thickness is sufficiently small that the remaining "ice" in the ice-free areas will have no impact on the results of the inversions and will avoid crashing the model. We will add a sentence to better explain this point.

C:

Rewritten  (lines 208-211)

- Eqn. 5, elsewhere - assume this is meant to be a centered dot indicating a dot product? (i.e., "dot(u,n) = 0")

A:

Yes, we will correct the text.

C:

Corrected (Eq.5, 7, 8)

- l. 266-268 - I understand what you're saying here, but it seems circular to say "our choice of input is consistent with our output (which somehow depends on the choice of input)" - maybe just use the reference to Meier?

A:

We understand what you mean, but in fact this is not totally circular. A few ice thicknesses from the boundaries the results should be insensitive to the details of the boundary condition (e.g. Mangeney et al., 1996, Gagliardini and Meyssonier, 2005), so the results in the interior can be used to justify that this is also a reasonable assumption at the boundaries. We will better clarify this point so that it does not appear as circular.

*Mangeney, F. Califano, O. Castelnau, Isothermal flow of an anisotropic ice sheet in the vicinity of an ice divide, J. Geophys. Res. 101 (12) (1996) 28,189–28,204.*

*Gagliardini, J. Meyssonnier, Lateral boundary conditions for a local anisotropic ice flow model, Ann. Glaciol. 35 (2002) 503–509.*

**C:**

Rewritten  (lines 279-283)

- l. 284 - might be good to include references for this statement.

**A:**

The following references will be added : Jay-Allemand et al. (2011), Gillet-Chaulet et al. (2012); Larour et al. (2014); Shapero et al. (2016); Maier et al. (2021)

*Jay-Allemand, M. et al. (2011) 'Investigating changes in basal conditions of Variegated Glacier prior to and during its 1982–1983 surge', The Cryosphere, 5(3), pp. 659–672.*

*Gillet-Chaulet, F. et al. (2012) 'Greenland ice sheet contribution to sea-level rise from a new-generation ice-sheet model', Cryosphere, 6(6), pp. 1561–1576.*

*Shapero, Daniel R., et al. 2016. Basal resistance for three of the largest Greenland outlet glaciers. Journal of Geophysical Research - Earth Surface 121(1): 168–180.*

*Maier, N. et al. (2021) 'Basal traction mainly dictated by hard-bed physics over grounded regions of Greenland', The Cryosphere Discussions, pp. 1–31.*

*Larour, E., Utke, J., Csatho, B., Schenk, A., Seroussi, H., Morlighem, M., et al. (2014). Inferred basal friction and surface mass balance of the Northeast Greenland Ice Stream using data assimilation of ICESat (Ice Cloud and land Elevation Satellite) surface altimetry and ISSM (Ice Sheet System Model). The Cryosphere, 8(6), 2335–2351.*

**C:**

References are added (lines 300-301)

- Fig. 3 - why not show the mismatch here, instead of in the appendix?

**A:**

We will add a subpanel in Figure 3 showing the mismatch.

**C:**

Subpanel is added in Fig.3

- l. 353 - what do you mean by "relatively" short distances?

**A:**

Here we mean that basal conditions can be heterogeneous even over distances of a few ice thicknesses (a few kilometers in our results). The sentence will be rewritten as : "as the basal conditions of this sector are heterogeneous and can likely change from an inferred hard to weak bed over distances of a few ice thicknesses."

**C:**

Rewritten  (lines 370-371)

- l. 368-372 - I'm not sure I completely understand these lines, and I think part of my confusion might come from calling the 24 datasets "time steps". By "restart from the optimal solution" do you mean that you use the parameters from the optimal solution as a starting point for each of the 24 datasets?

A:

Yes, "time steps" might be misleading. You understood correctly. We use the solution obtained for the MWS observations as a starting point for each of the 24 independent inversions corresponding to 24 velocity maps. This part will be rewritten as : "To reduce the computational burden for the monthly inversions, the basal friction coefficient field is initialised using the optimal solution obtained with the MWS observations. A new independent inversion is then run with the 24 data-sets using the same optimal value for the regularisation parameter"

C:

Rewritten  (lines 376-377, 387-389)

- l. 380 - why only the early half of each month? If there aren't significant differences between the early and late halves of each month, it would be good to mention that here.

A:

We kept only the early half of each month so that the size of the figure would remain reasonable. The mismatch for the early and late parts for the majority of months is similar.  We believe that showing only the early half of each month is sufficient to illustrate the average difference between the model and the observations and its trend over a year. We will add a statement that the second half of months usually shows similar mismatches to the first half.

C:

Rewritten (lines 399-300)

- Figures - it would be good to have some scale bars to help readers connect the text (e.g., "10-15 km from the margin") with the locations in the Figure

A:

Instead of scale bars, we use the uniform 10-km white grid on all figures from Fig.1 onwards. We will add the reminder about the grid size to all captions.

C:

All corresponding captions are updated

- l. 417 - how significant a change is this 2%? Would be good to have some idea of the variation here.

A:

Agreed. We will add the corresponding absolute values in the changes in speed for ud and us.

C:

Rewritten  (lines 436-437)

- l. 465-466 - this is an interesting observation - is there a phyiscal interpretation for why this might be the case?

A:

Yes, there is a physical interpretation of the hysteresis between sliding and basal friction. Similar observations or modeling results have been found in other studies. Sugiyama and Gudmunsson (2017) studied short-term variations in ice flow on an Alpine glacier in relation to subglacial water pressure. They showed that velocity was greater as pressure increased than as it decreased (for equivalent water pressures - consistent with a hysteresis). The increase in velocity with increasing pressure was interpreted as an opening of the subglacial water cavity and/or longitudinal stress coupling with the upper parts of the glacier. This linkage was also studied numerically by Iken (1981) by modeling basal slip on undulating bedrock as the water-filled cavities grew and shrank. Iken's modeling also predicted that a small drop in water pressure below steady-state values would result in a rapid decrease in slip rate. We will discuss this point in the revised version of the manuscript.

*Sugiyama, S. and Gudmundsson, G. H. (2004) 'Short-term variations in glacier flow controlled by subglacial water pressure at Lauteraargletscher, Bernese Alps, Switzerland', Journal of Glaciology, 50(170), pp. 353–362.*

*Iken, A. (1981) 'The effect of the subglacial water pressure on the sliding velocity of a glacier in an idealized numerical model.', Journal of Glaciology, 27(97), pp. 407–421.*

C:
Rewritten  (lines 493-494)

- l. 497 - could you (briefly) describe the differences between Eqn 12 and the "similar expression" proposed by Zoet and Iverson (2020)?

A:
Eqn 12 is directly the sliding law proposed by Gagliardini et al. (2009) for q=1 following the formulation from Schoof (2007). This sliding law has been developed for hard beds with cavitation. The form of the expression proposed by Zoet and Iverson (2020)  combines processes of hard-bedded sliding and bed deformation. While this friction law has additional parameters to account for the different physical processes controlling basal motion over till (i.e. till strength and clast size), the form of relationship between basal motion and friction ends up being quite similar to that over a hard bed. The main difference is that the effective pressure that appears in the denominator in Eq. 12 is not to the power n in Zoet and Iverson (2020). Because of this, for a given basal friction, the formula given by Zoet and Iverson predicts that the velocity should tend to 0 at high effective pressure, while Eq. 12 tends towards the velocity predicted by the Weertman friction law. We will explain this point in the revised version of the manuscript.

C:
Rewritten  (lines 523-530)

- l. 510 - this seems like it would be an issue?

A:
We assume that water pressure exceeding ice overburden is usually a very short-term event, thereby on the addressed "longer" timescale its neglect would not be a problem for the interpretation of the results. Additionally, from a technical point of view, both very small positive and negative N (e.g. 1e-5 and -1e-5) practically lead to the same outcome of near-zero friction in the model (which is unable to reproduce the other effects one might associate with water pressures exceeding overburden, such as hydraulic jacking).  Note that the water pressures greatly exceeding ice overburden is very unlikely; for instance, Doyle et al. (2015)

found that an unusual cyclonic late-summer rainfall generates maximum water pressure only of 100.5% of overburden.

*Doyle, S., Hubbard, A., van de Wal, R. et al. Amplified melt and flow of the Greenland ice sheet driven by late-summer cyclonic rainfall. Nature Geosci 8, 647–653 (2015)*

C:

Rewritten (lines 547-549)

- l. 536 - supplementary materials? I don't see Fig. S9 or Table S1.

  A:

  We are referring here to the supplementary materials of the cited paper Maier et al. (2021).

  C:

  No changes

- l. 634-636 - I'm not sure I understand what you're saying here.

  A:

  We are trying to explain that the total force balance during summer is still sufficient to prevent the glacier from collapsing (i.e. large-scale unstable sliding). Therefore, when friction locally becomes very small and the ice accelerates, the local change in stress is transmitted by longitudinal stress coupling to other places that will thereby offer enhanced flow resistance (larger friction) to maintain the global force balance.

  C:

  Rewritten

- l.638 - missing reference

  A:

  We will correct the text.

  C:

  Reference restored

- l. 718 - missing reference

  A:

  We will correct the text.

  C:

  Reference restored

- l. 731 - this is probably true for satellite observations, but ground-based radar interferometry potentially provides a way to do this (e.g., Caduff et al., 2015)

  A:

  Correct, we will change "remote" observation to "satellite".

  C:

  Rewritten (line 775)

**Reviewer #2**

- Line 1: I found this opening sentence a little unclear on first reading, I think because modelling is implied by "better constraints" but never explicitly mentioned. I'd suggest something structured more like "Due to increasing surface melting …, better constraints on … are required by models".
  **A:**
  Agreed.
  **C:**
  Rewritten  (line 1-2)

- Line 5: I'd expand this to say "using the ice-flow model Elmer/Ice", as it's possible some readers may not have come across it before.
  **A:**
  Agreed.
  **C:**
  Rewritten  (line 5)

- Line 23-4: Is this the authors' own assumption, or are there other studies to cite?
  **A:**
  Indeed, there are many other studies that have linked water pressure and glacier acceleration during the melt season. The overviews cited in the previous line (Davison et al, 2019; Nienow et al., 2017) widely cover this topic. We will rewrite the sentence to avoid ambiguity.
  **C:**
  Rewritten  (line 23)

- Line 122-3: It would be useful to add a brief explanation of the "master and slave" terminology.
  **A:**
  These terms will be changed to "primary" and "secondary" as requested by reviewer #1.
  **C:**
  Rewritten  (line 123)

- Line 123: It would be good to specify what x and y are, since this is the first mention of them (presumably polar stereographic north as in fig. 1?)
  **A:**
  Agreed.
  **C:**
  Rewritten  (line 124)

- Line 126: State what LOWESS stands for.
  **A:**
  LOWESS stands for LOcally Weighted Scatterplot Smoothing (Cleveland, 1979; Cleveland and Devlen, 1988) and is also known as locally weighted polynomial regression.The acronym and source references will be given in the revised text.
  *Cleveland, W. S. (1979) 'Robust locally weighted regression and smoothing scatterplots',*
  *Journal of the American Statistical Association, 74(368), pp. 829–836.*

*Cleveland, W. S. and Devlin, S. J. (1988) 'Locally weighted regression: An approach to regression analysis by local fitting', Journal of the American Statistical Association, 83(403), pp. 596–610.*
**C:**
Rewritten (lines 127-129)

- Line 162: Assumed by who? Should be made clear if this is the authors' own assumption, or citing another reference.
  **A:**
  This is assumed by us. We will clarify this point. Note that except a short remark made by Berthier et al. (2005), to the best of our knowledge, no previous description of the impact of changing shadows length on ice speed measurements have been published.
  *Berthier, E. et al. (2005) 'Surface motion of mountain glaciers derived from satellite optical imagery', Remote Sensing of Environment, 95(1), pp. 14–28.*
  **C:**
  Rewritten (line 167)

- Line 163: $v_y$ is a velocity vector, not speed.
  **A:**
  Agreed.
  **C:**
  Rewritten (line 168)

- Line 204: A brief explanation/sentence on kriging could be useful.
  **A:**
  Kriging is a widely used technique of thickness interpolation in ice sheet mapping. To stay consistent in our text, we will add a very short note on the consequences of kriging usage for the topography data quality, but without explanations on the method. We will simply refer to Morlighem et al. (2017) who describes how kriging is applied in BedMachine Greenland.
  *Morlighem, M. et al. (2017) 'BedMachine v3: Complete Bed Topography and Ocean Bathymetry Mapping of Greenland From Multibeam Echo Sounding Combined With Mass Conservation', Geophysical Research Letters, 44(21), pp. 11,051-11,061.*
  **C:**
  Rewritten (lines 215-216)

- Line 210-11: There are several acronyms here which could be fully introduced. I'd certainly specify Digital Elevation Model and Advanced Very-High-Resolution Radiometer. Perhaps expanding the names of specific models isn't necessary, but ASTER and SPOT-5 should probably be given relevant citations.
  **A:**
  Agreed.
  **C:**
  Rewritten (lines 221-224)

- Line 267-8: It doesn't make sense to justify the initial condition using the results it produces, which is how this reads to me. This sentence should be reworded to make the meaning clear.

**A:**

This point has already been raised by the first reviewer. As explained in our reply, this is not totally circular as in diagnostic the results in the interior of the domain are insensitive to the details of the boundary condition. We will improve this discussion in the revised version of the manuscript.

**C:**

Rewritten (lines 279-283)

- Line 273: What is the reason for this choice of friction law? Given the later focus on interpreting results using an effective pressure-based law, I think an explanation is needed for why that law wasn't used in the inversions to begin with.

  **A:**

  Good point. We will better explain this in a revised version of the manuscript. We preferred to invert the effective friction coefficient (\beta in Eq. 8) and then interpret the temporal variations of \beta in terms of effective pressure in a second step. There are several reasons for this. First, it should be numerically more stable to use a linear relation, and, in winter, the effective pressure-based law is close to the Weertman regime and thus weakly sensitive to N, so the results would be much more sensitive to the regularisation terms. Second, as we show in the manuscript, this two-step approach allows us to discuss the choices that are made to calibrate the parameters (As and C) of the effective pressure-based law.

  **C:**

  Rewritten (lines 535-540)

- Fig 3: I think a fourth panel is needed here, showing the difference from observed velocities. The mismatch is currently discussed without a visual aid.

  **A:**

  Agreed, we will add a subpanel in Figure 3 showing the mismatch between observed and modelled surface speed. As this figure shows the results obtained for the winter mean state (January/February/March), the mismatch is very similar to that of the individual months of January, February or March. Thus, we initially expected that the Figure A1 in appendix would be sufficient.

  **C:**

  Subpanel added in Fig.3

- Line 332-3: State this more clearly: ud=us-ub.

  **A:**

  Agreed.

  **C:**

  Rewritten (line 350)

- Line 380-1: What is the reason for choosing only the early halves? Wouldn't it be useful to see the later halves as well, especially for the months where conditions change quickly?

  **A:**

  We show only one half of each month to keep the figure at a reasonable size. As the differences between the early and late halves are not large for the majority of months, we feel

only showing the early halves is enough to demonstrate the variability. This statement will be more clearly articulated in the text.

C:

Rewritten (lines 399-300)

- 417-19: Isn't the limited effect of deformation at least partly a result of the choice mentioned previously to neglect deformation profiles when setting up the model? This could be a misunderstanding on my part based on what was said in lines 267-9.

  A:

  It is possible that there is a misunderstanding coming from lines 267-9 of how the ice deformation is taken into account in our model inversions. We will better explain this point in the revised manuscript.

  In lines 267-9, we state that the deformation profile is not prescribed in the starting conditions of an initializing inversion (using the winter mean speed (WMS) data. With a full-Stokes model, the 3D velocity field will quickly adapt to the boundary conditions, first of all to the basal friction, thereby the deformation will appear progressively for the areas at a distance of few ice thicknesses from the lateral boundaries. In the interior of the domain, e.g. where we compare our deformation profiles with those measured by Maier et al. (2019) in Fig.5, we assume that 200 iterations is enough for the WMS-based inversion to converge and reproduce entirely the deformation which the ice column would have for the given conditions.

  As the 24 seasonal inversions start from this WMS-produced solution, their initial conditions already contain the vertical deformation. Here, as we don't change the surface topography over time, only model-inferred basal friction can influence the deformation profile, making it variable from one inversion to another. Here we discuss that this effect is small but still visible.

  C:

  Rewritten (lines 438-445)

- Fig 6: In the top row, it would be good to use the same scale on the y-axis in each case, and to include horizontal grid lines like in the bottom row.

  A:

  We will add the horizontal grid lines for the bottom row. Considering the y-axis in the top row, during the manuscript preparation we tested many options and found that the same scale is less optimal. While it makes the plots more easily comparable, the individual nuances of friction and pressure behavior are lost for points A and C.

  C:

  Horizontal grid is added

- Line 470-4: Can this behaviour be explained? The other two points behave as I'd expect (ie. an inverse relationship between speed and friction), but this stands out as more of an anomaly.

  A:

  Yes, this behaviour can be explained and is what is predicted by the Weertman friction law, i.e. friction in absence of cavitation. As described in lines 631-639, point B follows a power friction law (friction increases with sliding) most likely due to longitudinal stress-gradient

coupling with regions being much more actively forced by meltwater (like point A). Simply put, the friction rise here is a consequence of acceleration imposed from outside.

In detail, we interpret annual friction and ice speed changes at Point B as follows: from January to May, the gradual recharge of the subglacial water system locally reduces the friction and the velocity slowly increases (van der Wal et al. 2015, Harper et al. 2021). In May, when surface melting begins, local topography and/or organisation of the hydrological system does not lead to an increase in water pressure and consequent facilitation of sliding. However, via longitudinal coupling to other accelerating areas such as point A, point B is forced to accelerate as well. Higher sliding speed for a relatively unchanging set of bed properties leads to a higher local friction.

This type of "passive" melt season response has been inferred in numerous previous studies in Greenland (Ryser et al., 2014b, Price et al. 2008, Maier et al. 2021, Young et al. 2019)

*Ryser, C. et al. (2014) 'Caterpillar-like ice motion in the ablation zone of the Greenland ice sheet', Journal of Geophysical Research : Earth Surface, 119, pp. 2258–2271.*

*Maier, N. et al. (2021) 'Basal traction mainly dictated by hard-bed physics over grounded regions of Greenland', The Cryosphere Discussions, pp. 1–31.*

*Young, T. J. et al. (2019) 'Physical Conditions of Fast Glacier Flow: 3. Seasonally-Evolving Ice Deformation on Store Glacier, West Greenland', Journal of Geophysical Research: Earth Surface, 124(1), pp. 245–267.*

*Van De Wal, R. S. W. et al. (2015) 'Self-regulation of ice flow varies across the ablation area in south-west Greenland', Cryosphere, 9(2), pp. 603–611.*

*Harper, J., Meierbachtol, T., Humphrey, N., Saito, J., & Stansberry, A. (2021). Variability of Basal Meltwater Generation During Winter, Western Greenland Ice Sheet (preprint). Ice sheets/Greenland.*

*Price, S. F., Payne, A. J., Catania, G. A., & Neumann, T. A. (2008). Seasonal acceleration of inland ice via longitudinal coupling to marginal ice. Journal of Glaciology, 54(185), 213–219.*

C:
No change

- Line 621: Is there a reason for not showing a profile through point B? Since it displayed different behaviour from the other points, it could be interesting to see that here as well.
  A:
  The initial reason was to keep the number of figures reasonable. We will consider adding it to the Appendix.
  C:
  Figure A6 is added to the Appendix

- Line 634-6: It's not clear to me what is being said in this sentence. It needs rewording.
  A:
  Agreed. We are trying to explain that the total force balance during summer is still sufficient to prevent the glacier from collapsing. Therefore, when friction locally becomes very small and the ice accelerates, the local change in stress is transmitted by longitudinal stress coupling to other places that will thereby offer enhanced flow resistance (larger friction). We will rephrase accordingly.
  C:
  Rewritten (lines 672-678)

- Line 645-6: I think I understand the meaning here, but this sentence is unclear. Is it that conditions are the same/similar down the whole length of profile C'?
  A:
  The sentence will be rewritten as: "The whole 30km of profile C' mainly exhibits the same conditions as were described for point C. "
  C:
  Rewritten (lines 688-689)

- Fig A5(b): What are the dotted lines? Mention them in the caption.
  A:
  Caption will be updated.
  C:
  Caption update

- Technical corrections
  A:
  Thank you for the technical corrections, we will correct all of them.
  C:
  All requests are adopted

---

## Author Response (AR2)

Dear Dr.Horgan,
thank you a lot for the detailed proofreading and the list of corrections. All of them are integrated into the text.

==============================

Minor corrections:

L15 'constantly losing mass' to 'losing mass' (accuracy)

L19 'the mass loss…' to 'mass loss…' (definite article usage)

L30 'the ice albedo' to 'ice albedo' (definite article usage)

L32 'The surface runoff…' to 'Surface runoff…' (definite article usage)

L39 'the ice dynamics.' to 'ice dynamics.' (definite article usage)

L39 'computes the effective…' to 'computes effective…' (definite article usage)

L40 'relates the basal…' to 'relates basal…' (definite article usage)

L66 'the effective…' to 'effective…' (definite article usage)

L67 'As for the subglacial hydrology, because…' to 'Because…' (Concise language)

L70-71 'so that the models' to 'so models' (Concise language)

L75 'In real applications, models' to 'Models…' (Concise language)

L84 '…now includes..' to '…now include…'

L96 'At the same…by temporal' Awkward sentence, consider changing to 'Temporal variations of basal hydrology have been addressed by several studies using inferred…'

L98 'the effective pressure' to 'effective pressure' (definite article usage)

L98 'Thus, the main…' Non-sequitar, change to 'The main…'

L104 'the studied region' to 'the study region'

L113 'We address the questions of how' to 'We address how' (Concise language)

L117 'we conclude on…' to 'we conclude with'

L124 'east-wast' to 'east-west'

L129 'is applied on' to 'is applied to'

L130 'that they can' to 'that the product can' (remove ambiguity)
L132 'the noise' to 'noise', 'the spatial coverage' to 'spatial coverage'

L137 'extremes, thus' to 'extremes. Thus…' (run-on sentence)

L143 'estimation' to 'estimate'

L153 'speed magnitudes' to 'speed'

L162-163 'Therefore, the y-component…' Check the logic of this sentence. If the east-west component was very small also, then this would be true but not as currently stated.

L170 'suppose' to 'assume'

L171 'affected: the' to 'affected. The…' (run-on sentence)

L175 ', the ice flow' to ', ice flow'

L183-184 'Russell basin' to 'Russell Sector' for consistency

L184 'and infer it for the ' to 'infer the'

L189 'as well, the' to 'either. The' (run on sentence)

L193 'In this section' to 'In the remainder of this section…'

L216 '...on the bed…' to 'in the bed…'

L221 'As for the…, they are….' to 'The...are…' (concise language).

L247 'done in...' to 'from...'

L251 ', certainly' 'This is likely…' (run-on sentence and exactness of language.)

L264 'E: to validate.' to 'E. To validate' (run-on sentence).

Table 1 caption. 'The list of…' to 'List of….'

L270 'distances higher' to 'distances greater'

L280 'deformation of ice column occurring further across the inland...' to 'deformation of the

ice column occurring in the inland…'

L281 'Along with that, we…' to 'We…'

L281 'results few' to 'results a few'

L293 'norm of of'

L296 'choice of the friction law' to 'choice of friction law'

L300 'Inverting the' to 'Inverting for the'

L324 'For that, the' to 'The…' (ambiguity)

L327 'Based on that, ' remove ambiguity.

L336 'is generally below 5%' be precise 'is below X%' or 'is below 5% in XX% of cases'

L345 ', certainly because…' to 'likely because' (exact language).

L345 'of the friction' to 'of friction'

L346 'could be the underestimation' to 'could be due to underestimation'

L350 'increase up to' (which would mean 40-30--45-65) or 'increase to up to'?

L367 ', being higher on…' to ', it being higher on…'

L388 'be-monthly' 'twice-montly' (bi-monthly is ambiguous)

L397 '...with the initial…' to 'when compared with the initial…'

L397 'stagnates' word choice, perhaps 'stabilises'

L403-404 '...velocity maps are indeed the most important for these periods, often exceeding' to 'velocity, often exceeds…' (most important how?)

L405 'remains' to 'is'

L429-430 Concatenate single line paragraph to next paragraph.

L436 'and absolute' to 'and an absolute'

L441 '"deformation"' to 'deformation'

L487 'clear correlation' to 'clear anti-correlation between friction and sliding speed'

L492 'Such effect,' to 'Such an effect,'
L518 'without cavity' to 'without cavities' or 'without cavitation'

L529 'provide good fit' to 'provide a good fit'

L535 'to invert the' to 'to invert for the'

L540 'law, what is described further' to 'law.' Concise language

L568 'Properly saying, the..' to 'The…'

L570 'basal melting water' to 'basal melt water'

L571 'inferred at the mesh…' to 'at the mesh..'

L575 'It also' Our ASW estimate also has…' (remove ambiguity)

L578 'with a low' to 'with low…'

L585 'would certainly lead' to 'would lead'

L586 'is the uncertainties on Asw estimated as' 'is the uncertainty in Asw estimated by…, computed as:'

L603 'would be not valid' to 'would be invalid'

L604 'that it is' 'than otherwise'

L605 'we stay limited' to 'we are limited'

L607-610. This long run-on sentence is hard to follow. Break it up and clarify.

L640 '...as from the defined boundaries for Eq. 13 CN is always positive, meaning..' to 'as Eq. 13 shows CN is always positive, meaning…'

L641 '...margin, similarly to the….stat...' to '...margin. This is similar to the modelling of the water routing system in steady state…, suggesting an increased...'

L650 '...to conclude from our inversions on' to 'from our inversions to draw conclusions on

L655 'winter almost half of inferred by us.' to 'winter that are almost half those inferred by us.'

L658 'Fig. A5' to 'Fig. A6'

L659 'two profiles' to 'three profiles'
L661 'it shows' ambiguous, clarify.

L663 'in the Weertman law' to 'in Weertman's law..'

L671 'picks' to 'peaks'

L682 'it would be' to 'this is…'

L682 'by the basal melting water' to 'by basal melt water'

L686 'is probably a mix between point A…..adjusting….' to 'is probably a mix between point A, where local variations of water pressure occur, and point B, which responds to non-local longitudinal stresses.' or similar.

L691 ', we probably approach a Coulomb-like behavior where…' to ', Coulomb-like behavior is likely approached, where…'

L693 'consistent with important cavitations' important how? 'consistent with cavitation, and…'

L726 'since some moment the' to 'the…' either remove or be precise.

L736 'and in' to 'and'

L746 'we can conclude' to 'we conclude'

L749 'the water pressure at the base' to 'basal water pressure'.

L751 'at a much higher…' quantify 'that can now be obtained at a time resolution of ….allowing… This offers the…

L766 '"ridges" would correspond…' to 'while the ridges correspond..'

L768 concatenate small paragraph to previous paragraph.

L771 'would still be needed' 'is still needed…'

L773 concatenate small paragraph to previous paragraph.

L785 'appear small' to 'are small'

L803 Include GIMP data source.